# Taming I2V models for Image HOI Editing:
# A Cognitive Benchmark and Agentic Self-Correcting Framework

**Jiayi Gao** [1]  **Qingchao Chen** [2]  **Yuxin Peng** [1]  **Yang Liu** [1]

## Abstract

Current image editing methods excels at static attributes but fails at **complex Human-Object Interactions (HOI)**, a critical challenge unaddressed by existing benchmarks that conflate HOI with static attributes, relying on global metrics incapable of simultaneously assessing dynamic interaction validity and entangled human-object pair preservation. Thus, we first introduce **HOI-Edit**, a comprehensive benchmark with three progressive cognitive levels, which features an automated metric **HOI-Eval** that first reliably evaluates instance-level interaction by letting VLM Q&A after thinking with images containing grounded Human-Object pair. Considering the task's essence of *remodeling dynamic relationships*, we benchmark Image-to-Video (I2V) models, finding them inherently suited for dynamic editing due to their temporal generation capabilities. Crucially, beyond superior performance, this capability provides a "replay of the failure process", offering unique diagnosability into *why* errors occur. We thus propose **SCPE (Self-Correcting Process Editing)**, a novel, agentic self-correcting framework that constrains the generation of I2V models through iteratively refined prompts, enabling the generated videos to more accurately present the target HOI. Extracted frames from these videos are the final editing results. On HOI-Edit, SCPE achieves performance competitive with state-of-the-art (SOTA) editing models like Nano Banana on interaction. Code is available at https://github.com/oceanflowlab/HOI-Edit.

[1]Wangxuan Institute of Computer Technology, Peking University, Beijing, China [2]National Institute of Health Data Science, Peking University, Beijing, China. Correspondence to: Yang Liu <yangliu@pku.edu.cn>.

*Proceedings of the $43^{rd}$ International Conference on Machine Learning*, Seoul, South Korea. PMLR 306, 2026. Copyright 2026 by the author(s).

## 1. Introduction

Instruction-based image editing (Brooks et al., 2023; Zhang et al., 2023; Feng et al., 2025; Yu et al., 2025; Zhang et al., 2025c) has witnessed remarkable progress in modifying static attributes, yet **Human-Object Interaction (HOI)** editing presents a significant challenge. Unlike generic editing, HOI editing necessitates a **rational verb-driven pairwise editing**. The core challenge lies in a strict **contextual constraint**: instead of freely hallucinating new targets, the model must **synthesize a contextually logical interaction** involving the **pre-existing subject and object anchored in the scene**. This prevents the model from bypassing identity preservation by regenerating entities, thereby serving as a rigorous test of **maintaining scene context consistency**—a fundamental prerequisite for building robust World Models.

However, advancing this field is hindered by two critical gaps. **First, no dedicated benchmark exists to evaluate this task**. Given the inherent complexity of HOI, robust evaluation must transcend **foundational editing**; it demands assessing the model's capacity to **understand scene context for precise interaction grounding**, and to reason about the underlying **causal preconditions and physical consequences**. **Second, standard evaluation protocols are insufficient**. Reliance on global metrics (e.g., CLIP-score(Radford et al., 2021)) or isolated entity checks struggle to discern specific targets or focus on pair-wise context. This highlights an urgent need for **entangled pair-wise region-sensitive metrics** to explicitly evaluate the consistency and rationality of interacting human-object pairs.

To comprehensively evaluate both entity preservation and dynamic process rationality, we introduce **HOI-Edit**, a benchmark structured around three hierarchical cognitive levels to assess editing capabilities across *foundational state transitions*, *scene context understanding*, and *causal reasoning* (shown in Fig. 2). Specifically, **L1: Foundational Edits** targets the shift from *Static Appearances to Dynamic Interactions*, e.g., editing "holding" to "riding" a skateboard requires transforming human posture and object position while preserving identity features like facial and texture details. **L2: Context Spatial Understanding** evaluates the model's grasp of spatial relationships for initial entity selection (e.g., "topmost apple") and terminal state place-

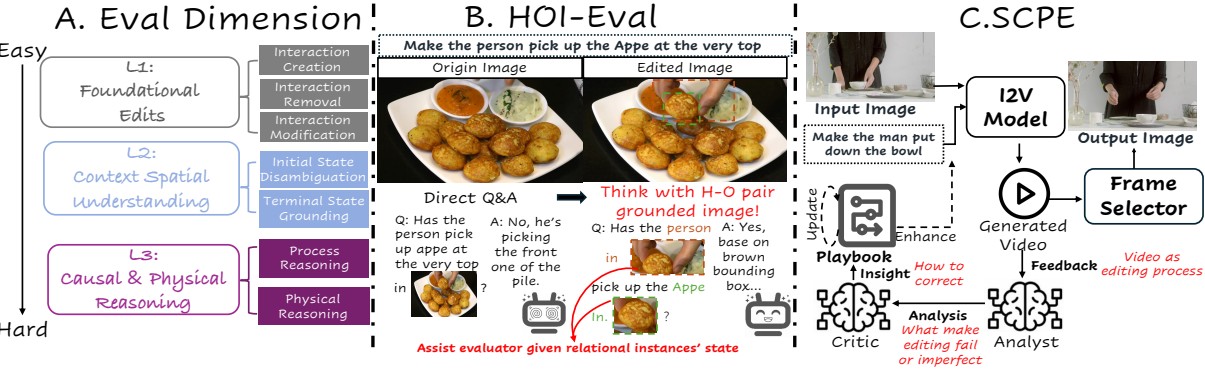

*Figure 1.* **Overview.** We present (A) **HOI-Edit**, the first benchmark forHOI editing across 3 cognitive levels; (B) **HOI-Eval**, a novel metric for verifying HOI correctness; and (C) **SCPE**, an agentic framework optimizing I2V models' HOI editing ability.

ment (e.g., "into the vase"), ensuring dynamic interactions align precisely with spatial descriptions. **L3: Causal and Physical Reasoning** requires simulating event causal chains with strict logical and physical plausibility. It must reflect the unstated procedural sub-actions implied in instructions (e.g., 'stirring' food in a pot needs 'removing the lid' first) in edited outputs, and infer physical visual changes, including illumination effects and non-rigid material deformation (e.g., object distortion during cutting).

To address the limitations of global metrics in capturing specific relations, we propose **HOI-Eval**, a grounded evaluation protocol based on the "Thinking with Pair-wise Regions" paradigm. By explicitly annotating paired bounding boxes for interacting subjects and objects, HOI-Eval offers three key advantages: (1) **Robust Identity Preservation**: Bounding boxes disentangle multiple instances from the image context, enabling isolation of target subjects/objects for strict **pair-wise identity** consistency evaluation. (2) **Elimination of Referential Ambiguity**: Annotations act as visual anchors to distinguish targets from similar objects, ensuring the model focuses on the intended entity. (3) **Context-Aware Interaction Reasoning**: By locating subject-object pairs alongside global context observation, HOI-Eval empowers VLMs to precisely assess **interaction status changes**, as well as their **contextual spatial accuracy and logical & physical rationality**

Using HOI-Edit, we benchmark a comprehensive set of editing models. Given the task focus on re-modeling dynamic relationships, we extend evaluation to Image-to-Video (I2V) models, leveraging their intrinsic temporal generation capabilities for dynamic reconstruction remodeling (Gao et al., 2025). Empirically, video-based models show significant advantages in the open-source domain. Crucially, this supe-

riority includes **diagnosability**: unlike static image models that only exhibit spatial artifacts (indicating *what is wrong*), failed I2V generations provide a temporal "replay of the failure process" (revealing *why it failed*), e.g., a hand trajectory erroneously steering towards a wrong target. To exploit this diagnosability, we propose **Agentic SCPE** (Self-Correcting Process Editing). As shown in Fig 1(C), SCPE relies on a dynamic 'Playbook'—**a structured knowledge base mapping failure patterns (e.g., physics violations) to validated prompting strategies**. In this loop, a 'Video Analyst' conducts **sample-wise error analysis**, enabling a 'Critic' to synthesize **global insights**, **incrementally update** the Playbook, and iteratively refine naive instructions into robust video prompts. This approach achieves SOTA performance across all metrics among open-source models and even outperforms commercial SOTA model Nano Banana(Google, 2025) on Interaction scores.

In summary, our contributions are threefold: First, we introduce **HOI-Edit**, the first multi-level benchmark designed to evaluate the shift from static attribute editing to dynamic HOI editing—a key capability of *world simulation*. Second, we propose **HOI-Eval**. Rooted in the **"Thinking with Pair-wise Regions"** paradigm, it leverages spatial anchors to enable **region-sensitive evaluation of entangled human-object pairs** for strict relational verification. Finally, we present SCPE, an agentic playbook-driven framework for instruction refinement. SCPE enhances existing I2V model's capability for HOI editing, achieving performance comparable to commercial SOTA baselines.

## 2. Related Work

**Instruction-based image editing.** The visual generation fields (Xie et al., 2025; Wang et al., 2024; Xu et al., 2025a)

has evolved from static diffusion models (e.g., FLUX.1 Kontext(Esser et al., 2024), Qwen-Image-Edit(Wu et al., 2025a)) to unified frameworks like Bagel(Chang et al., 2025) and UniWorld(Min et al., 2023), with commercial systems like Nano Banana(Google, 2025) achieving state-of-the-art performance. However, while these models excel at style transfer or object replacement, they predominantly formulate editing as a static pixel repainting task. They treat interactions as instantaneous attributes, overlooking that Human-Object Interaction (HOI) (Xu et al., 2025c) is inherently a continuous temporal process (e.g., eating snacks implies the prerequisite motion of picking them up). Although ChronoEdit(Wu et al., 2025b) validates video priors for temporal consistency, it targets general video editing and overlooks the specific intricacies of HOI dynamics. We diverge by introducing SCPE, the first agentic(Zhang et al., 2025b) framework to 'tame' I2V models with low cost. Uniquely, SCPE enables adaptive correction on generated dynamic frame chains, synthesizing error experiences to derive generalized optimization strategies for process-aware HOI editing.

**Image Editing Benchmarking.** Evaluation metrics have evolved from global statistics (Radford et al., 2021; Zhuoying Li & Liu, 2025) to "VLM-as-a-Judge" protocols (Pu et al., 2025), yet current benchmarks(Han et al., 2025; Pan et al., 2025) suffer from a structural gap in interaction validity. First, datasets such as I2EBench (Ma et al., 2024) and ImgEdit (Ye et al., 2025) focus primarily on single-object attributes or static conversation memory. While frameworks like ByteMorph(Chang et al., 2025) have started addressing dynamic editing aspects, they lack fine-grained protocols for evaluating entangled pairwise interactions and the strict identity preservation required for interaction modifications. Critically, these methods frame editing as a flat task, ignoring the complexity of interaction demands that span a progressive cognitive hierarchy—from basic dynamic verbs and precise spatial control to context- and physics-aware causal reasoning. Thus, effective evaluation must assess not only interaction occurrence but also its accuracy and rationality. To fill this gap, we propose the first benchmark tailored for human-object interaction validity across three hierarchical cognitive levels.

## 3. HOIEdit Dataset

### 3.1. Hierarchical Cognitive Evaluation

We introduce **HOI-Edit**, a benchmark assessing HOI editing across three progressive cognitive tiers: **Foundational Editing (L1)**, **Context Spatial Understanding (L2)**, and **Causal and Physical Reasoning (L3)**, as shown in Fig 2.

**L1: Foundational Edits** evaluates the ability to modify the **interaction** between a specific human-object pair. The eval-

uation criteria are twofold: (1) **Relation Transformation**, ensuring the successful creation, removal, or modification of the interaction; and (2) **Entity Preservation**, ensuring the generative process strictly adheres to the **human and object IDs** present in the original image.

**L2: Context Spatial Understanding.** L2 evaluates spatial comprehension including **initial entity selection** and **terminal state placement**. The challenge is to establish precise entity-to-location mappings within the scene. (1) **Initial State Disambiguation** tests identifying and editing a **specific target** among similar candidates based on spatial context (e.g., "pick up the appe *at the very top*"). (2) **Terminal State Grounding** focuses on the **post-editing configuration**, requiring control over the entity's **final destination**. This ranges from **fine-grained local placement** (e.g., "put flower *into vase*") to **long-range spatial repositioning** (e.g., "person walk to window to draw curtain").

**L3: Causal and Physical Reasoning.** L3 assesses the high-level ability to infer **information** unmentioned in the instruction but necessary for reasonable simulation. This involves decomposing processes and adhering to physical laws. (1) **Process Reasoning** requires constructing logical causal chains, such as deducing unstated **prerequisite steps** (e.g., "remove lid" before "stir") or identifying **implicit tools** necessary for the goal (e.g., 'open the lid' before stirring food ). (2) **Physical Reasoning** demands adherence to rigorous physical constraints, specifically **illumination consistency** (i.e., Is the surrounding darken?) and **irreversible non-rigid deformations** (i.e. Is the carrot cut into pieces).

### 3.2. Data Curation

We curated high-resolution frames from HOI video datasets (Liu et al., 2025a) and conducted rigorous manual annotation targeting three objectives to enable fine-grained evaluation, as shown in Fig. 3(top): (1) **Context-Aware Instruction Design**: We manually authored instructions tailored to the specific scene context. This manual curation ensures a high diversity of interactions (see Fig. 5) while strictly verifying that the interactions are **contextually plausible** and that all involved entities **physically exist** within the scene. For instructions in L2 and L3, we respectively incorporate spatial cues for localization and omit details for process or physical effects to compel reasoning. (2) **Systematic Region Annotation**: To facilitate precise **spatial region-wise evaluation**, we systematically annotated bounding boxes for the interacting subject $B_h$ and object $B_o$. Furthermore, to support higher-level reasoning, we defined **auxiliary context regions** $B_{aux}$: for example, procedural entity regions for L3 (e.g., intermediate tools). This structured spatial grounding set $\mathcal{B} = \{B_h, B_o, B_{aux}\}$ serves as the foundation for our focused **grounding-based Q&A evaluation**. (3) **Multi-Faceted Question Construction**: Fi-

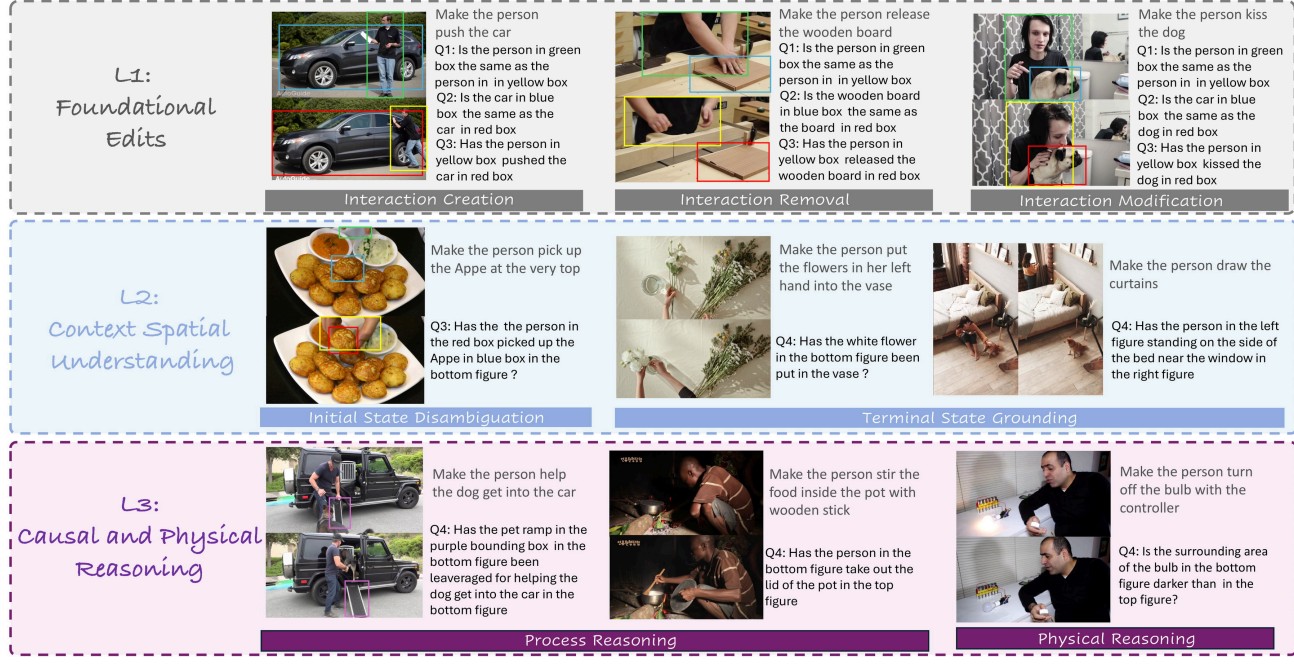

*Figure 2.* **Illustrative examples for hierarchical cognitive level in *HOIEdit*.**

nally, we designed a comprehensive grounding-based Q&A suite tailored to distinct evaluation needs: First, **to assess core pair-wise subject-object interactions**, we constructed two **pair-wise identity questions** to strictly verify subject and object retention, respectively(e.g., Q1-L1, Q2-L1) and one **interaction status question** to confirm the action occurrence(e.g., Q3-L1). Second, **for spatial understanding and reasoning (L2/L3)**, we tailored **context-aware questions** to verify terminal spatial states for L2(e.g., Q-L2:"*Is the stick in hand?*") and intermediate procedural steps(e.g., Q-L3: "*Was the file used?*") or **resulting physical phenomena** for L3. The final dataset comprises 357 L1, 202 L2, and 146 L3 samples across diverse categories (Fig. 4).

## 3.3. HOI-Eval Metric

We propose **HOI-Eval**, a three-step metrics to access edited HOI, as shown in Fig.3(bottom). **Step 1: Entity Localization and Association.** To enable region-sensitive evaluation, we establish a precise correspondence between the annotated ground truth boxes $\mathcal{B} = \{B_h, B_o, B_{aux}\}$ in original image $I$ and the target entities in the edited image $\hat{I}$. Treating the pair $(I, \hat{I})$ as a **pseudo-video sequence**, we employ tracking (Ravi et al., 2024) to propagate masks from $\mathcal{B}$ to $\hat{I}$, yielding the associated regions $\hat{\mathcal{B}} = \{\hat{B}_h, \hat{B}_o, \hat{B}_{aux}\}$. We fall back to prompt-based detection (Liu et al., 2024) only when tracking fails, ensuring robust **region association**. For rare samples with severe interaction-induced deformation (e.g., chopping), we adaptively switch to global semantic

validation. **Step 2: Identity Verification (H, O).** To strictly verify identity, we crop and stitch the ground truth box $B_h$ from $I$ and its tracked counterpart $\hat{B}_h$ from $\hat{I}$, with tag "Original image" and "Edited image" overlaid (see Fig. 3) to assist VLM discrimination. Based on it, we ask **pair-wsie ID similarity questions**(Q1-L1, Q2-L1) to compute **Human (H)** and **Object (O)** consistency scores (0-1). This explicitly focuses the model on specific entities to effectively detect **identity drift**. **Step 3: Interaction and Rationality Evaluation.** With the tagged images, we first verify interaction by prompting the VLM for a confidence as interaction score **(I)** (0-1) with interaction status questions. To validate adherence to spatial, logical, and physical constraints, we ask context-aware questions for corresponding information (e.g. Q-L2 and Q-L3, utilizing relative positions for L2 and intermediate tools or processes for L3 when available) to assess rationality via binary judgments. To penalize illogical edits, we introduce the **I+Q&A** metric: it retains the Interaction score ($I$) only if the context-aware question is answered correctly, otherwise setting it to 0. It reflects task success under constraints, not just perceptual similarity. All verifications are performed by Gemini 2.5 Pro (Google, 2025), Detail prompts for HOI-Eval are in appendix C .

## 4. Pilot Experiments

To identify bottlenecks in HOI editing, we conducted a pilot benchmark using the HOI-Eval against different methods: open-source image editing models (Flux.1 Kontext(Esser

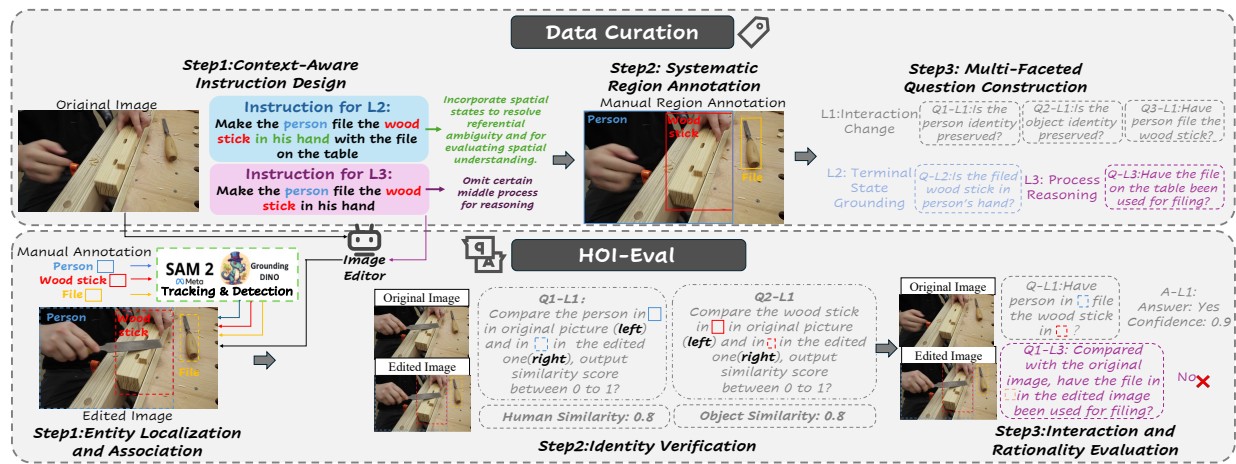

*Figure 3.* **Overview of benchmark construction and evaluation pipeline.**

*Table 1.* **Quantitative Comparison.** We report Interaction (I), Human (H), and Object (O) consistency across three cognitive levels. Note that **I+Q&A** sets the interaction score to 0 if the answer for context-aware question is wrong. SCPE only update for 1 iteration.

| Method | Source | L1: Foundational | | | L2: Understanding | | | | L3: Reasoning | | | |
|---|---|---|---|---|---|---|---|---|---|---|---|---|
| | | I↑ | H↑ | O↑ | I↑ | I+Q&A↑ | H↑ | O↑ | I↑ | I+Q&A↑ | H↑ | O↑ |
| Flux.1 Kontext (Esser et al., 2024) | Open | 0.4961 | 0.8991 | 0.5347 | 0.5192 | 0.2780 | 0.8998 | 0.4631 | 0.2052 | 0.0423 | 0.9653 | 0.8777 |
| ByteMorph (Chang et al., 2025) | Open | 0.4469 | 0.2000 | 0.2220 | 0.4279 | 0.2240 | 0.2227 | 0.1757 | 0.4156 | 0.1571 | 0.7480 | 0.6584 |
| Step1X-Edit(Liu et al., 2025b) | Open | 0.5396 | 0.7985 | 0.6533 | 0.5159 | 0.4058 | 0.7997 | 0.6472 | 0.5874 | 0.4194 | 0.8446 | 0.7640 |
| ChronoEdit(Wu et al., 2025b) | Open | 0.5823 | 0.7418 | 0.6023 | 0.5574 | 0.4608 | 0.7515 | 0.6160 | 0.5620 | 0.4345 | 0.8205 | 0.7359 |
| Bagel(Deng et al., 2025) | Open | 0.6326 | 0.7940 | 0.4790 | 0.6065 | 0.4781 | 0.7804 | 0.5030 | 0.6013 | 0.4061 | 0.8516 | 0.5701 |
| Qwen-Image-Edit PLUS(Wu et al., 2025a) | Closed | 0.6128 | 0.9343 | **0.8775** | 0.5984 | 0.4928 | 0.9395 | 0.7924 | 0.5878 | 0.3870 | 0.9593 | 0.8602 |
| Nano Banana(Google, 2025) | Closed | 0.7271 | **0.9537** | 0.8609 | 0.7040 | 0.5960 | **0.9590** | 0.7706 | 0.7399 | 0.5782 | **0.9743** | **0.9185** |
| Wan 2.2 I2V (Wan et al., 2025) | Open | 0.6908 | 0.9166 | 0.7823 | 0.6608 | 0.5526 | 0.9113 | 0.7272 | 0.6822 | 0.5343 | 0.9306 | 0.8511 |
| Wan 2.2 I2V + SCPE | Open | **0.8423** | 0.9260 | 0.8640 | **0.7909** | **0.6952** | 0.9269 | **0.8260** | **0.8053** | **0.6528** | 0.9518 | 0.9073 |

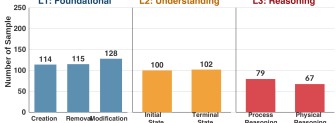

*Figure 4.* HOI-Edit data distribution

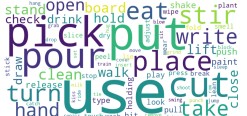

*Figure 5.* Interaction verbs

fail to model interaction dynamics. Consequently, their failure cases map manifest as imprecise instruction adherence in L1 and L2, illogical object or procedural artifacts (e.g., the appearance of a new scarf,armrest box and spoon in L1(bottom) and L3(middle & bottom)), physical violations (e.g., wrong reflections for a dog in L3(top)).

et al., 2024), ByteMorph(Chang et al., 2025), Step1X-Edit(Liu et al., 2025b), ChronoEdit(Wu et al., 2025b) with temporal reasoning, Bagel(Deng et al., 2025)); commercial static editors (Qwen-Image-Edit PLUS(Wu et al., 2025b), Nano Banana(Google, 2025)); and the Wan 2.2 I2V 14B(Wan et al., 2025). All experiments were conducted on single NVIDIA H20 GPUs.

**Image-based Approaches** As shown in Table 1, results reveal a distinct performance hierarchy. Open-source models exhibit suboptimal interaction performance across all levels, indicating a fundamental deficiency in modeling dynamic Human-Object Interactions. While commercial SOTA models like Nano Banana remain relatively robust and achieve high scores, qualitative analysis (Fig. 6) reveals their limitations. Since static paradigms lack temporal context, they

**I2V Model's Performance.** Given that HOI editing requires dynamic relationship remodeling, we benchmarked the image-to-video (I2V) baseline model Wan2.2. Experimental results show that Wan2.2 outperforms the state-of-the-art open-source image editing baselines (e.g., Bagel(Deng et al., 2025)) by a significant margin, thanks to its superior temporal generation capability. However, the SOTA model Nano Banana(Google, 2025) boasts excellent stability and maintains high interaction scores across all evaluation levels. In contrast, Wan2.2 fails to fully satisfy complex task constraints—such as subpar performance in drastic displacement interactions (e.g., failing to move the person in L2(top)) or weak reasoning ability regarding intermediate operational states (e.g., not open the armrest box before cleaning it in L3(middle)).

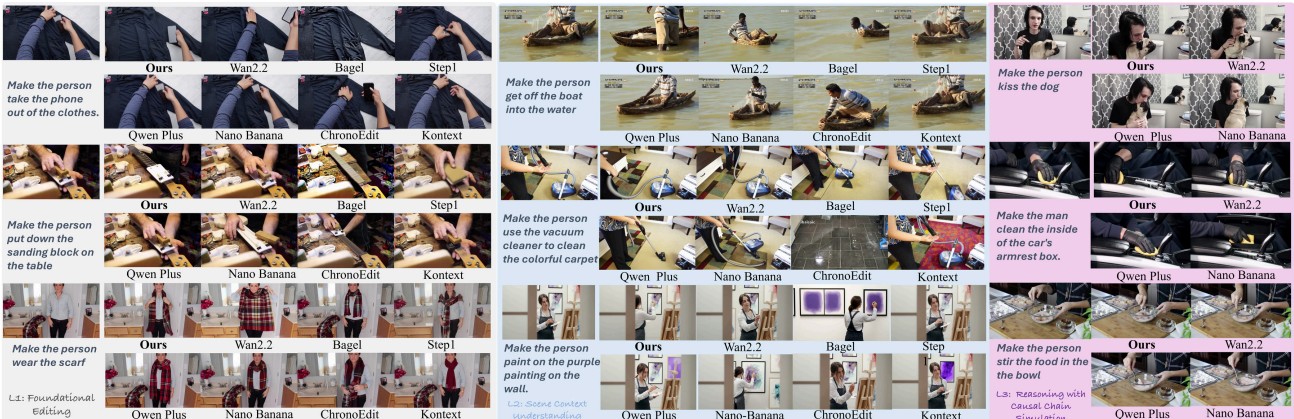

Figure 6. **Qualitative comparison on HOI-Edit.** where 'Qwen Plus' stands for Qwen-Image-Edit PLus

**I2V Model's Potential.** Interestingly, despite the performance gap, we observe a unique advantage in I2V models over static baselines. Unlike the opaque, irreversible artifacts in static editing (which only reveal *what* went wrong), I2V outputs provide a **"replay of the execution process."** which exposes *why* the failure occurred—for instance, capturing a hand *erroneously steers toward the nearest target*. This transforms the failure from a black-box error into a diagnosable event, raising a pivotal question: *Can such explicit procedural visual cues be leveraged to guide the model toward self-correction and precise improvement?*

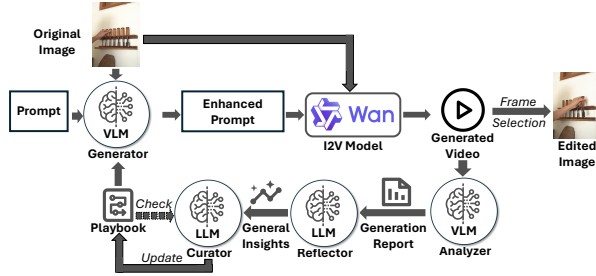

Figure 8. **Pipeline of SCPE.**

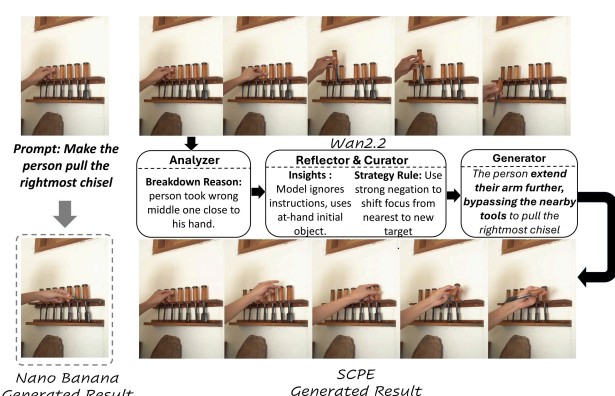

Figure 7. **Visualization for prompt enhance process.**

## 5. Self-Correcting Process Editing

To answer this question affirmatively, we propose **SCPE (Self-Correcting Process Editing)**. Unified by the backbone of Gemini 2.5 Pro (Comanici et al., 2025), SCPE operates as a closed-loop system where specialized agents leverage the model's multimodal capabilities (for Generator/Analyzer) or textual reasoning (for Reflector/Curator) to iteratively optimize generation. As illustrated in Figure 8, we demonstrate this workflow using the task of retrieving a

specific rightmost chisel as a running example: The **Generator** first queries the **Playbook**—a dynamic, initially empty knowledge base evolving via iterations—and leverages this knowledge to integrate the initial instruction with the input image, producing an **enhanced instruction** that guides the I2V model to precisely execute the target HOI editing; the generated video (top frames) shows suboptimal performance, with the hand grabbing the nearby middle chisel. Next, the **Analyzer** samples video frames and generates a **Generation Report** to diagnose the error: *"person took the wrong nearby middle chisel"*. The **Reflector** then processes this report to extract **General Insights**, framing the failure as a common pitfall: *"Model prioritizes nearest objects over Playbook guidance; strong negation is required to refocus on the target"*. Finally, the **Curator** uses these insights to incrementally **update** the Playbook; the Generator then leverages the updated Playbook and a refined prompt (e.g., *"Extend your arm further, bypass nearby tools to pull the rightmost chisel"*) to re-guide the I2V model, producing a corrected video where the rightmost chisel is retrieved.

**The Dynamic Playbook.** The Playbook serves as the system's evolving memory, mapping identified failure patterns to verified prompting strategies. As shown in Fig.7, It incorporates three components: (1) **Strategies and Hard Rules**, which encode high-level principles. For example, for com-

posite actions like "hammer the nail," the system learns to decompose the instruction into concrete sub-steps like "tool preparation" and "action execution" . (2) **Troubleshooting and Pitfalls**, functioning as an "immune system" built from error logs. For instance, if the Analyst detects a floating object, the Playbook records a rule requiring explicit surface contact for placement tasks. (3) **Reusable Templates**, which store verified prompt structures to resolve ambiguities, e.g., using relative spatial descriptors (e.g., "the rightmost one") to correct proximity bias in target selection.

**Frame Selection.** To extract the optimal result, we sample 15 frames from the generated video. With the model (Comanici et al., 2025) understanding multiple frames, we identify the single frame that best aligns with the editing instruction. Details about SCPE are in appendix B.

## 6. Experiments

### 6.1. Main Results

**Quantitative Results.** As shown in Table 1, SCPE achieves SOTA performance in interaction precision and identity preservation across nearly all cognitive levels. It also outperforms commercial SOTA Nano Banana(Google, 2025) in interaction metrics across all cognitive levels, demonstrating stronger instruction adherence, spatial understanding, and causal reasoning. Its slightly lower identity (H/O) scores result from Nano Banana's static architecture (optimized for original subject preservation) and the drastic spatial reconfigurations in I2V generation (may inducing entity movement/deformation). Besides, static models exhibit editing inertia: they often hallucinate new objects instead of modifying existing ones (e.g., L1 (bottom), L3 (middle & bottom) in Figure 6), and this *"failure to edit"* preserves original entities, yielding higher identity scores. Compared to original Wan 2.2, SCPE improves across all metrics, with superior instruction adherence for target interactions and robust identity preservation. Notably, substantial gains in the I+Q&A metric demonstrate enhanced spatial grounding, logical and physical reasoning, confirming that agentic-based process supervision and enhancement framework comprehensively elevates HOI editing abilities of video generation models.

**Qualitative Results.** Figure 6 is organized by cognitive complexity for comparison: **1) L1: Overcoming Hallucination & Inertia .** The left column shows two baseline failure modes: in "take the phone out" (top-left), baselines exhibit editing inertia (unchanged image); in "wear the scarf" (bottom-left), static models generate unrealistic artifacts ( hallucinations) to mimic prompts. SCPE overcomes both, executing removal/addition actions while preserving subject identity. **2) L2: Precise Spatial Grounding.** For "paint on the purple painting" (bottom-middle), vanilla Wan 2.2 misaligns the action to the *foreground easel*, and other base-

lines hallucinate global style changes; SCPE accurately navigates to the background canvas, resolving spatial ambiguity and rectifying the "limited spatial reachability" issue via long-range displacement. **3) L3: Physical Consistency & Procedural Reasoning.** In "kiss the dog" (top), SCPE maintains perspective consistency in complex environments (e.g., mirrors), while Nano Banana produces mismatched reflections (e.g., mirror dog faces away). SCPE also exhibits superior procedural reasoning: for "clean the inside of the armrest box" (middle), it is the only model that correctly executes the prerequisite step (opening the box) for cleaning, verifying grasp of causal dependencies and addressing baselines' "procedural inconsistency".

*Table 2.* **Correlation with Human Judgment.** We report the Pearson correlation ($Pr$) and statistical significance ($Pp$).

| Method | Subject (H) | | Object (O) | | Interaction | |
|---|---|---|---|---|---|---|
| | **P**$r$ | **P**$p$ | **P**$r$ | **P**$p$ | **P**$r$ | **P**$p$ |
| DINOv2 | 0.035 | 0.835 | 0.172 | 0.301 | - | - |
| CLIP | 0.253 | 0.125 | -0.027 | 0.872 | - | - |
| **HOIEval (Ours)** | **0.43** | **.007** | **0.47** | **.003** | **0.60** | **.002** |

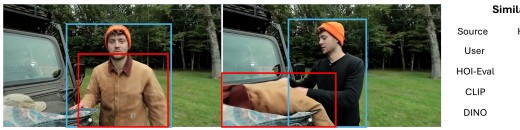

*Figure 9.* Subject-object similarity comparison: HOI-Eval vs global metrics (instruction: place jacket on car hood).

*Table 3.* **Ablation study. I** is interaction score and IDS is identity score (avergae of **H** & **O**).

| Method Variant | I ↑ | IDS ↑ |
|---|---|---|
| Wan 2.2 I2V 14B | 0.6804 | 0.8494 |
| + OPE | 0.7028 | 0.7385 |
| wo Playbook | 0.7625 | 0.8786 |
| **+ SCPE** | **0.8199** | **0.8954** |

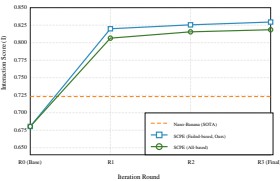

*Figure 10.* Iteration Analysis.

### 6.2. Ablation Study

**Metric Reliability and User Study.** To verify HOI-Eval metrics' reliability, we analyzed correlations between its automated metrics (interaction editing, identity preservation) and human evaluations (Table 2). Traditional global metrics (DINO(Zhang et al., 2022), CLIP(Radford et al., 2021)) mismatch human perception, showing near-zero or negative object consistency correlations; in contrast, HOI-Eval achieves significant cross-dimensional correlations, demonstrating better suitability for identity preservation in dynamic editing. Visualization in Fig. 9 confirms its strong alignment with human evaluations (Pearson correlation = 0.60) and robust identity recognition under dynamic

*Table 4.* **Quantitative Comparison on SCPE on different variants with inference time.** SCPE only updates for one iteration.

| Method | Time (min) | Overall | | |
|---|---|---|---|---|
| | | I↑ | H↑ | O↑ |
| ChronoEdit(Wu et al., 2025b) | 20.0 | 0.5709 | 0.7607 | 0.6337 |
| Nano Banana(Google, 2025) | - | 0.7231 | 0.9595 | 0.8469 |
| + SCPE (Ours) | - | 0.7310 | **0.9602** | 0.8351 |
| Wan 2.2 I2V(Wan et al., 2025) | 30.0 | 0.6804 | 0.9189 | 0.7807 |
| + SCPE (Ours) | 30.0 | **0.8199** | 0.9267 | **0.8565** |
| TurboDiffusion(Zhang et al., 2025a) | 1.4 | 0.6290 | 0.8890 | 0.6967 |
| + SCPE (Ours) | 1.4 | 0.7536 | 0.9118 | 0.7735 |

deformation (e.g., non-rigid jackets). On the 200-annotated-sample HICO-DET dataset(Chao et al., 2018; Lei et al., 2025), HOI-Eval reaches 98.5% interaction judgment accuracy, further validating its reliability and capability to assess complex dynamic interaction. Details of human evaluation protocols and HICO-DET experiments are provided in the appendix C.

**Ablation for SCPE Components.** To dissect SCPE's performance gains, we compared Wan 2.2's Official Prompt Enhancer (OPE) with a variant using only current-sample analysis (wo Playbook) (Table 3), yielding three key findings: **(1) Limitations of OPE:** OPE yields marginal I improvement but severe IDS degradation (0.8529 → 0.7313). This decline arises from OPE's "blind prediction" nature: lacking an understanding of how subject/object appearances evolve during dynamic interactions, it generates redundant and erroneous visual descriptions, leading to severe identity drift. **(2) Effectiveness of Visual Feedback:** In contrast, the Analyst-only variant (wo Playbook) uses visual feedback to dynamically enhance prompts based on generator's outputs. This rectifies stochastic physical distortions and delivers substantial performance gains, validating the correction mechanism. **(3) Impact of the Playbook:** The full SCPE framework achieves the highest I, confirming reflective memory's core value in distilling generalized rules and enabling cross-sample experience transfer.

### 6.3. Analysis and Discussion

**Effectiveness of Iterative Refinement.** Fig. 10 shows that SCPE converges rapidly, surpassing Nano-Banana at R1 after one playbook update. Although subsequent iterations (R2-R3) bring additional gains, the improvements are marginal while latency increases linearly. Therefore, **we adopt the R1 setting for the main experiments as the optimal efficiency-performance trade-off**. Moreover, the failed-based update strategy (Blue) consistently outperforms the all-based update strategy (Green), validating that deriving insights from analyzed failure cases yields more effective and robust refinements than indiscriminate updates.

**Why Video Priors Matter: Unlocking Process Correc-**

**tion.** As shown in Table 4, commercial SOTA Nano Banana(Google, 2025) also benefits from SCPE's refined instructions (demonstrating strong generalization for complex prompts), but its performance gain is far less significant than that of the I2V model. We attribute this gap to a core difference in diagnosability: static models like Nano Banana compress interaction causal chains into a single frame, so failures only appear as final spatial artifacts (revealing *what* is wrong), making logical breakdowns hard to pinpoint. In contrast, I2V models unfold temporal dynamics, providing a transparent "process replay" that exposes *why* failures occur. SCPE's in-process design leverages these temporal cues to intercept errors at their source, transforming editing from blind trial-and-error to logical process correction.

**Evaluator-method Coupling.** To rule out potential evaluation bias caused by using the same visual language model on both the method side and the evaluator side, we further introduce an independent evaluator, GPT-5.4(Singh et al., 2025), to re-evaluate the edited results in similar ways. The relative ranking of major methods, especially for interactionness, remains consistent under GPT-5.4 evaluation, as shown in Table 5. This indicates that the evaluation trend does not depend on a specific VLM evaluator. We also validate the agent-side flexibility of SCPE by replacing the proprietary Gemini reasoning engine with an open-source Qwen 3.5-VL agent(Team, 2026). As shown in Table 6, Qwen-based Playbook correction still brings consistent improvements over the Wan 2.2 I2V baseline across all three cognitive levels. These improvements are close to those achieved by Gemini-based Playbook correction, demonstrating that SCPE does not rely on a specific VLM reasoning backbone. Moreover, it shows that SCPE can be implemented with open-source models and still surpass closed-source models such as Nano-Banana. In addition, when Gemini 2.5 Pro is used only as an official prompt enhancement engine, the resulting improvements are much weaker and even harm identity preservation. This further distinguishes SCPE from generic VLM prompt rewriting. Overall, the independent evaluation with GPT-5.4 and the substitution experiment with Qwen 3.5-VL jointly demonstrate that the main performance gains come from the structured video-feedback mechanism and Playbook-based experience transfer, rather than from the inherent capability advantage or self-preference of a particular VLM.

**Efficiency and Scalability Analysis. (1) Gradient-free Framework with High Adaptability.** SCPE is a gradient-free framework, which makes it backbone-agnostic: it allows us to seamlessly switch to stronger video backbones (e.g., Wan 2.2) or acceleration methods (e.g., TurboDiffusion(Zhang et al., 2025a)) and instantly benefit from their capability gains, whereas the cost of optimizing the Playbook is negligible compared to ChronoEdit(Wu et al., 2025b)'s requirement of optimizing all DiT parameters of Wan 2.1 I2V with extensive data. **(2) Scalable Inference Efficiency.**

*Table 5.* Quantitative comparison on HOI-Edit benchmark. Models and results are processed under independent evaluator GPT-5.4.

| Method | Source | L1 (Foundational) | | | L2 (Understanding) | | | L3 (Reasoning) | | |
|---|---|---|---|---|---|---|---|---|---|---|
| | | I↑ | H↑ | O↑ | I+Q&A↑ | H↑ | O↑ | I+Q&A↑ | H↑ | O↑ |
| Nano Banana | Closed | 0.7346 | **0.9437** | **0.9195** | 0.6134 | **0.9391** | 0.881 | 0.5587 | **0.9437** | **0.9319** |
| Wan 2.2 | Open | 0.7325 | 0.8865 | 0.8804 | 0.6104 | 0.8978 | 0.8033 | 0.5012 | 0.904 | 0.8832 |
| Wan 2.2 + SCPE | Open | **0.7870** | 0.8999 | 0.8967 | **0.6595** | 0.913 | **0.8928** | **0.5878** | 0.9218 | 0.9066 |

*Table 6.* **Evaluation of VLM-agnostic flexibility.** This comparison isolates the Playbook's logic from the specific reasoning engine, comparing Gemini and Qwen 3.5-VL. The near-consistent gains indicate that the accumulated physical priors, rather than the inherent strength of a specific VLM, are the primary drivers of dynamic editing performance.

| Method | Source | L1 (Foundational) | | | L2 (Understanding) | | | | L3 (Reasoning) | | | |
|---|---|---|---|---|---|---|---|---|---|---|---|---|
| | | I↑ | H↑ | O↑ | I↑ | I+Q&A↑ | H↑ | O↑ | I↑ | I+Q&A↑ | H↑ | O↑ |
| Wan 2.2 I2V (Wan et al., 2025) | Open | 0.6908 | 0.9166 | 0.7823 | 0.6608 | 0.5526 | 0.9113 | 0.7272 | 0.6822 | 0.5343 | 0.9306 | 0.8511 |
| Wan 2.2 I2V + SCPE | Open | **0.8423** | **0.9260** | **0.8640** | 0.7909 | **0.6952** | **0.9269** | **0.8260** | **0.8053** | **0.6528** | 0.9518 | **0.9073** |
| Wan 2.2 I2V + Qwen Playbook | Open | 0.8368 | 0.9115 | 0.8427 | **0.7922** | 0.6857 | 0.9048 | 0.7724 | 0.7928 | 0.6454 | **0.9551** | 0.8835 |
| Wan 2.2 I2V + Gemini OPE | Open | 0.7396 | 0.8287 | 0.6658 | 0.7194 | 0.5910 | 0.8115 | 0.6429 | 0.6988 | 0.5724 | 0.8326 | 0.7013 |

By integrating SCPE with faster I2V model TurboDiffusion, we achieve SOTA open-source performance in just 1.4 minutes—significantly outpacing ChronoEdit with temporal reasoning as well and surpassing Nano Banana in interaction quality. This confirms SCPE's future-proof scalability: despite current video generation latency, our framework naturally adapts to faster backbones, ensuring high-speed HOI editing as underlying models accelerate.

*Figure 11.* Visualization for Playbook

**How Playbook enhance the instruction.** Fig11 shows representative Playbook entries targeting the dominant failure modes across 3 cognitive levels. The mechanism operates as a "diagnosis-and-cure" system: for each level, the agent identifies *Pitfalls*—such as static optimization bias (L1), spatial ambiguity (L2), or causal shortcuts (L3)—and retrieves corresponding *Strategies* to impose constraints. These strategies transform abstract instructions into structured directives, enforcing dynamic identity binding, physical checks, and causal chaining to effectively guide the video model.

**Generalization on Other Dynamic Benchmark** To further validate robustness, we evaluated SCPE on the ByteMorph Benchmark (Chang et al., 2025). As detailed in Table 7, SCPE outperforms the native ByteMorph model (fine-tuned with 6M data) in both *Human Motion* and *Interaction* categories. This confirms the general advantage of integrating video backbones with SCPE for dynamic editing tasks.

*Table 7.* **Evaluation Results Comparison on ByteMorph.**

| Method | Category | CLIP-text | GPT Score |
|---|---|---|---|
| ByteMorph (Chang et al., 2025) | HM | 0.3094 | 93.80 |
| | I | 0.3101 | 89.86 |
| **Wan I2V + SCPE (Ours)** | HM | **0.3225** | **95.12** |
| | I | **0.3249** | **93.97** |

# 7. Conclusion

In this paper, we argue that robust HOI editing requires a paradigm shift from static pixel in-painting to dynamic process simulation. To validate this, we introduced the HOI-Edit benchmark and the SCPE framework, which "tames" I2V models to correct interaction failures iteratively. Our results demonstrate that leveraging temporal video priors allows for superior reasoning in spatial and causal contexts. Ultimately, this work provides a rigorous testbed and a viable pathway that approximate physical and causal processes at the interaction level, a prerequisite for learned world models rather than a full physical simulator.

## Impact Statement

This paper advances the field of machine learning by tackling the challenge of dynamic Human-Object Interaction (HOI) editing in image-to-video models. Our work lays a critical prerequisite for learned world models—rather than serving as a full physical simulator—and holds promising implications for applications like intelligent content creation, virtual scene simulation, and human-computer interaction system optimization.

## Acknowledgements.

This work was supported by the grants from the National Natural Science Foundation of China (62372014, 62525201, 62132001, 62432001), Beijing Nova Program and Beijing Natural Science Foundation (4252040, L247006).

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

## A. Appendix Overview

This Appendix provides comprehensive technical details and experimental analysis to support the reproducibility of the SCPE framework and the HOI-Edit benchmark. Specifically, Section B discloses the core system prompts for the four specialized agents (Generator, Analyzer, Reflector, and Curator) , the Dynamic Playbook's delta-update mechanism , and the rigorous Semantic Frame Selection protocol used to identify the definitive interaction moment. Subsequently, Section C details the HOI-Eval methodology, including the (1) Comparison VQA prompt templates, (2)the scoring rubrics for identity persistence and visual quality, and (3) quantitative evidence for the reliability of our SAM 2 based regional association alongside human correlation study designs. Section D provides extended fine-grained performance analysis and qualitative visualizations. Finally, Section E provides a candid discussion of current limitations, such as inference latency, background preservation stability and the data scale.

## B. SCPE Implementation Details

In this section, we provide the specific implementation prompt details of the SCPE framework.

### B.1. Agent System Prompts

To ensure reproducibility, we present the core system instructions used for the four specialized agents. In the actual implementation, these prompts are dynamically populated with user inputs and retrieved Playbook entries, and multimodal inputs (images/videos) are attached as media parts to the LLM API.

---

**Role 1: Generator Agent**

**System Instruction:** You are a video prompt expert specializing in Human-Object Interaction (HOI). Your task is to combine an input image, a simple instruction, and a "Playbook" manual to generate a detailed prompt for video generation.
**Input Context:**

- **[Input Image]:** The provided source image (attached).

- **[Instruction]:** "{instruction}" (Simple user command).

- **[Playbook]:** A JSON structure containing Strategies, Pitfalls, and Templates.

**Task Execution Steps:**

1. **Visual Analysis:** Analyze the objects, human subjects, and spatial relationships within the input image.

2. **Intent Analysis:** Decipher the **core intent** of the simple instruction.

3. **Playbook Adherence:** Strictly follow the "Strategies", avoid the "Pitfalls", and utilize the "Templates" defined in the Playbook.

4. **Prompt Generation:** Generate a **detailed, time-sequenced, and visually faithful** video prompt.

**Output Schema (JSON):** `{ "reasoning": "string", "enhanced_prompt": "string" }`

---

**Role 2: Analyzer Agent**

**System Instruction:** You are a strict video analyst. Your task is to determine whether the generated video successfully completes the designated Human-Object Interaction (HOI) task.
**Input Context:**

- **[Original Image]:** (Attached) The source image used for video generation.

- **[Original Instruction]:** "{instruction}" (The initial user command).

- **[Enhanced Prompt]:** "{enhanced_prompt}" (The detailed prompt produced by the Generator).

- **[Generated Video]:** (Attached) The dynamic result produced by the I2V model.

**Task Execution Steps:**

---

1. **Dynamic Intent & Visual Consistency:** Analyze whether the generated video faithfully executes the **core dynamic intent** of both the original instruction and the enhanced prompt, while maintaining visual consistency with the original image.

2. **Motion Rationality:** Given the first-frame scene, ensure the entire motion process is reasonable and coherent.

3. **Background Stability:** Ensure the camera viewpoint and background remain completely stationary (no camera movement or background changes).

4. **Criterion Examples:**
   - If the instruction is "pick up the cup" (dynamic) but the video is a static image of "holding a cup," it is a **failure**.
   - If the instruction specifies a "red cup" but the video shows a blue one, it is a **failure**.

**Output Schema (JSON):** `{ "reasoning": "string", "success": boolean, "critique": "string" }`

---

### Role 3: Reflector Agent

**System Instruction:** You are a senior HOI analyst. Your job is to distill "General Wisdom" from a failed generation prompt and a specific failure analysis of the generated video.
**Background:** A "Generator" produces a video based on a potentially defective prompt, and an "Evaluation Model" analyzes the generated video and provides a specific failure report.
**Input Context:**

- **[Failed Prompt]:** The enhanced prompt used for the failed generation.

- **[Analysis Report]:** The Evaluation Model's JSON output (e.g., `{success, critique}`).

**Task Requirements:**

1. Read the failed prompt and the specific failure report from the Evaluation Model (e.g., "The cup in the video is blue instead of red as in the original image" or "The person did not move").

2. **Abstract:** Identify the universal principle underlying this specific failure (e.g., "The prompt failed to enforce visual feature consistency with the input image").

3. **Constraint:** Your output must NOT contain specific object names, colors, or scene descriptions (e.g., avoid mentioning "cup", "red", "man"). It must be generalized rules applicable to future unseen tasks.

4. Output two core elements: a "root_cause" (the fundamental reason for the failure) and a "key_insight" (actionable guidance for future optimization).

**Output Schema (JSON):** `{ "root_cause": "string", "key_insight": "string" }`
**Example:**

- Specific Failure: "The cup in the video is blue, but it is red in the original image."

- Abstracted Output (JSON): `{ "root_cause": "The prompt failed to enforce visual features from the input image.", "key_insight": "When visual consistency is critical, strong guiding terms (e.g., 'the exact same as in the image') must be used in the prompt to modify objects." }`

---

### Role 4: Curator Agent

**System Instruction:** You are the curator of the knowledge base. Your role is to maintain a *Playbook* containing only high-quality, non-redundant entries. The Playbook consists of three components: **strategies**, **templates**, and **pitfalls**.
**Input Context:**
- **[Insight]:** JSON containing `root_cause` and `key_insight` from the Reflector.

- **[Current Playbook]:** The current Playbook state (JSON).

**Initial Bootstrap Playbook:** We initialize the system with a small seed Playbook before the first Curator call. These seed entries serve as example rules to help the Curator understand the expected format and level of abstraction of Playbook knowledge. They are also included in [Current Playbook] during training so that newly proposed entries can be checked against them. The default seed entries are:

- **strategies:** `[strategy-001]:` `The video prompt must describe the start, process, and end of the action in detail.`

- **templates:** `[template-001]:` `A {viewpoint} shot in which {person}'s {body part} approaches {object} from {direction}...`

- **pitfalls:** `[pitfall-001]:` `Pitfall: using vague verbs (e.g., "interact"). Remedy: concretize them into specific actions such as "push", "pull", or "turn".`

**Task Execution Steps:**

1. Read the `key_insight` in [Insight], which is a generalized optimization suggestion derived from failure analysis.

2. **Redundancy Check:** Carefully review [Current Playbook]—including the bootstrap entries above—and determine whether the `key_insight` already exists in any of the three components in any form.

3. **Perform Actions:**
   - If the insight is new:
     - a. Rewrite it into a concise and actionable Playbook entry;
     - b. Determine which component it belongs to (`strategies`, `templates`, or `pitfalls`);
     - c. Create an incremental update operation of `type`: "ADD".
   - If the insight is redundant:
     - a. Return an empty `operations` list to avoid introducing redundant content.

4. **Core Constraint:** Your primary responsibility is to prevent context collapse. Therefore, only delta updates are allowed, and you are prohibited from rewriting the entire Playbook.

**Output Schema (JSON):** `{ "reasoning": "string", "operations": [ { "type": "ADD", "section": "strategies" | "templates" | "pitfalls", "content": "string" } ] }`

## B.2. Dynamic Playbook Mechanism

The Playbook is implemented as a structured JSON file, serving as the evolving long-term memory of the system. To ensure stability, we implement a **Delta Update** mechanism. The *Curator* agent outputs a list of operations (e.g., ADD). Subsequently, a deterministic merging function (`merge_playbook`) implements the addition of new content while strictly preventing redundancy.

## B.3. Video Generation Configuration

The underlying Image-to-Video (I2V) generation is powered by the Wan2.2-i2v 14B model. For all experiments, we maintain a consistent generation resolution of $1280 \times 720$ (720P) with a fixed duration of 5 seconds.

## B.4. Frame Selection Prompt Details

To extract the final editing result from the generated video sequence, we do not rely on random sampling. Instead, we employ a **two-stage Semantic Frame Selector** powered by a Visual Language Model (VLM), specifically Gemini 2.5 Pro. For each generated clip, we uniformly sample $N$=15 frames and encode them as a temporally ordered image sequence, rather than using native video input. The selector returns a single key-frame index, and the corresponding frame is saved as the final editing result.

**Stage I: Completion Query Formulation.** In the first stage, the VLM receives only the user instruction, without access to the video frames. It infers the interaction type as either `static` or `dynamic`, and formulates a binary completion question that can be used to judge whether the intended action has been completed. For example, given the instruction "Make the person put down the cup", the model may generate the question: "Has the person put down the cup?" This formulation avoids vague "best-frame" selection and encourages the later selector to identify the earliest frame where the interaction outcome is fully established.

**Stage II: Key-Frame Selection.** In the second stage, the VLM receives the original instruction, the generated completion question, the action-type-dependent sharpness rule, and the sampled frames indexed from 1 to $N$. The selector first operates in `strict_earliest_yes` mode, where it must return the smallest frame index that can be answered "Yes" to the completion question while satisfying the visual sharpness requirement. For static actions, the selected frame should be clear and blur-free; for dynamic actions, slight motion blur is allowed if the interaction remains visually interpretable. If no frame satisfies the strict criterion, the selector switches to `best_effort` mode and returns the frame that is most aligned with the instruction, closest to completion, and has visible subject/object content with acceptable visual quality. We do not default to the last frame.

---

**System Prompt: Two-stage Semantic Frame Selector**

**Stage I: Completion Query Formulation**
Given only the user instruction, infer:

- **Interaction Type:** `static` or `dynamic`.

- **Completion Question:** A binary Yes/No question that checks whether the intended action has been completed.

The completion question should test the final interaction outcome rather than ask which frame is visually best. Avoid vague wording such as "best frame" or "most beautiful frame". The question should support the earliest-completion policy in Stage II.
**Stage II: Key-Frame Selection**
You are given:

- **User Instruction:** [Insert Instruction, e.g., "Make the person put down the cup"]

- **Completion Question:** [Generated in Stage I]

- **Interaction Type:** `static` or `dynamic`

- **Frame Sequence:** Frames indexed from 1 to $N$

**Selection Protocol:**

- **Strict mode (`strict_earliest_yes`):** Select the smallest frame index where the completion question can be answered "Yes" and the frame satisfies the corresponding sharpness rule.

- **Static actions:** The subject and object should be clear and free from noticeable motion blur.

- **Dynamic actions:** Slight motion blur is acceptable if the interaction is still clearly recognizable.

- **Fallback mode (`best_effort`):** If no frame meets the strict criterion, still return one frame index. Choose the frame most aligned with the instruction, closest to completion, with visible subject/object and acceptable visual quality.

- Do not select a mid-action frame when a completed outcome is available.

- Do not default to the last frame.

**Output Schema:** { "selected_frame_index": integer, "strict_pass": boolean, "selection_mode": "strict_earliest_yes" or "best_effort", "match_score": float, "reasoning": "string" }

---

## C. HOI-Eval Methodology Details

In this section, we provide the exact prompts and scoring rubrics used in our automated evaluation pipeline (HOI-Eval).

### C.1. Interaction Scoring

For the final interaction score, we employ a "Comparison VQA" strategy. We provide the model with both the original and the edited images, where the latter is augmented with detected bounding boxes to guide the spatial reasoning.

---

**Prompt 3: Comparison VQA for Interaction**

**System Instruction:**
You will receive TWO tagged images in chronological order: (1) **Original Image**, (2) **Edited Image**. The edited image features blue (subject/person) and red (object) bounding boxes showing the DETECTED positions for interaction. Answer the following question based on these images to justify if the interaction happens.
**Input:**

- **Original Image**

- **Edited Image**

- **Question**:Manual Designed for Interaction Veritification

**Criteria:**

- **Focus:** Concentrate on the subject and object within the bounding boxes in the edited image.

- **Relationship:** Check if the relationship between subject and object visually suggest the interaction.

- **Evidence:** The interaction must be visually evident and an improvement/change over the "Before" state.

**Output Schema:** Provide a strict JSON response. The `"confidence"` score (0.0 to 1.0) reflects certainty in a "yes" answer. If `"Interaction_Answer"` is "no", `"confidence"` must be 0.0.

```
{
"Interaction_Answer":  "yes",
"confidence":  0.95
}
```

---

## C.2. Identity Scoring

A key contribution of HOI-Eval is the decoupling of **Identity Persistence** and **Visual Quality**. By incorporating the *Edit Instruction* as a semantic anchor, the VLM distinguishes between intended motion-induced changes and generation failures. The final **Similarity Score** is mathematically defined as the product of the two sub-scores ($S = I \times Q$), ensuring a "veto" effect where low visual realism significantly penalizes overall performance.

---

**Prompt 2: Entity Similarity & Quality Evaluation**

**Task:** Compare the tagged Original Image and the Edited Image. Identify the target entity and evaluate its consistency based on the *Edit Instruction* context.
**Input:**

- **Original Image**

- **Edited Image**

- **Question**:Designed for Human and Object Similarity

**Rubric A: Subject (Human) Evaluation**

- **Identity Score (0.0-1.0):** Focus on core semantic attributes.

  - **Compatibility:** Accept *Generative Infilling* (e.g., newly generated facial features during a turn) if the skin tone, build, and clothing match the original partial view.
  - **Identity Loss (<0.3):** Direct contradictions in gender, race, or clothes.

- **Quality Score (0.0-1.0):** Anatomic & Visual Realism **[CRITICAL]**

  - **Strict Penalty (<0.4):** Immediately penalize "Uncanny Valley" artifacts (waxy skin, dead eyes, asymmetry...), distorted extremities (extra joints, sausage fingers...), or a "pasted" appearance where lighting contradicts the background.

**Rubric B: Object Evaluation**

- **Identity Score (0.0-1.0):** Focus on instance persistence.

---

> – **Dynamic Tolerance:** Do not penalize changes in state (open/closed), location (held vs. falling), or occlusion (released) if the semantic identity of the object remains clear.
>
> • **Quality Score (0.0-1.0):** Geometric Integrity & Physics **[CRITICAL]**
>
> > – **Strict Penalty (<0.4):** Penalize if object's structure collapse (e.g., a cylinder becoming a flat box) or if rigid materials (e.g., a metal pot) appear "melty" or soft.
>
> **Final Metric Calculation:** `Subject & Object Similarity = Identity_Score × Quality_Score`

## C.3. Context-aware Question Verification

> **Prompt 3: Comparison VQA for Interaction**
>
> **System Instruction:** You will receive TWO tagged images in chronological order:(1) **Original Image**, (2) **Edited Image**. The edited image is annotated with specific bounding boxes: **blue** for the subject, **red** for the object, and **yellow** for auxiliary context regions.
> Your task is to answer the interaction question by analyzing the state change and spatial relations between these boxes.
> **Input:**
>
> • **Original Image**
>
> • **Edited Image**
>
> • **Context-Aware Question**
>
> **Criteria:** Analyze the state change in the Edited Image relative to the Original Image. Utilize the yellow box (if present, otherwise refer to regions described in the question) to assess spatial, logical, or physical constraints (e.g., surface contact or instrument usage) to provide a factual answer for context-Aware question.
> **Output Schema:** {
> ```
> "answer":  "yes",
> "reasoning":  "Briefly explain the interaction based on the positions of the blue,
> red, and yellow boxes."
> }
> ```

## C.4. Reliability of Region Association in HOI-Eval

To evaluate the robustness of the automated localization module (Step 1) in HOI-Eval, we conducted a comprehensive human audit on 390 samples generated by **ChronoEdit** (Wu et al., 2025b) across the entire benchmark. ChronoEdit was selected due to its representative median-level performance. We manually annotated the ground-truth bounding boxes for both the subject (human) and the object in the edited videos to provide a gold standard. Quantitative analysis demonstrates that our SAM 2-based temporal propagation method achieves a **93.7% tracking success rate** ($IoU > 0.5$). This high precision validates the reliability of our localization module in handling interaction-induced motion.

## C.5. Details for User Study

To further validate the reliability of our proposed **HOI-Eval** metric and the effectiveness of the **SCPE** framework from a human perspective, we conducted a comprehensive user study detailing the experimental design and statistical results. We randomly sampled 100 cases from the **HOI-Edit** benchmark, covering all three cognitive levels: L1 (Foundational), L2 (Spatial), and L3 (Causal). For each case, we presented 20 participants with the original image, the editing instruction, and the final frames generated by seven representative models: **Nano Banana**(Google, 2025), **Wan 2.2 I2V**, **Wan 2.2 + SCPE (Ours)**, **ChronoEdit**(Wu et al., 2025b), **Qwen-Image-Edit Plus**(Wu et al., 2025a), **ByteMorph**(Chang et al., 2025), and **Flux Kontext**(Esser et al., 2024). Participants were asked to rate the results on a 5-point Likert scale across two dimensions: (1) *Interaction Success*, where evaluators provided a comprehensive score determining if the edited image faithfully executed the instruction, strictly considering the absence of sudden extraneous objects, adherence to positional requirements, and logical/physical rationality; and (2) *Identity Preservation*, assessing how well the visual features of the original human subject and object were maintained. To quantify the alignment between automated assessment and human perception, we calculated both the Pearson and Spearman correlation coefficients between the human ratings and the scores assigned by Gemini 2.5 pro (Google, 2025).

## C.6. Details of HOI-Eval Reliability Validation on HICO-Det

To further validate the reliability of HOI-Eval in interaction judgment, we construct a 200-sample evaluation subset from HICO-DET. For each sample, we visualize the ground-truth human and object regions using blue and red bounding boxes, respectively, and ask the evaluator a yes/no question, such as: "Is the person in the blue bounding box eating the apple in the red bounding box?" The subset contains 100 positive samples with valid human-object interactions and 100 negative samples from the no-interaction category, where the questions are designed according to the corresponding task-relevant interaction categories.

Using Gemini 2.5 Pro as the evaluator, HOI-Eval achieves 98.5% interaction judgment accuracy on this balanced subset, further validating its reliability in assessing fine-grained human-object interaction semantics.

# D. Detailed Performance and More

## D.1. Detailed Performance on Sub-dimensions in Three Cognitive Levels

As illustrated in the radar chart (Figure 12), our proposed **Wan+SCPE** demonstrates a dominant performance envelope across all seven fine-grained cognitive sub-dimensions, confirming that the agentic self-correction mechanism effectively bridges the gap between simple pixel-level manipulation and complex logical interaction synthesis.

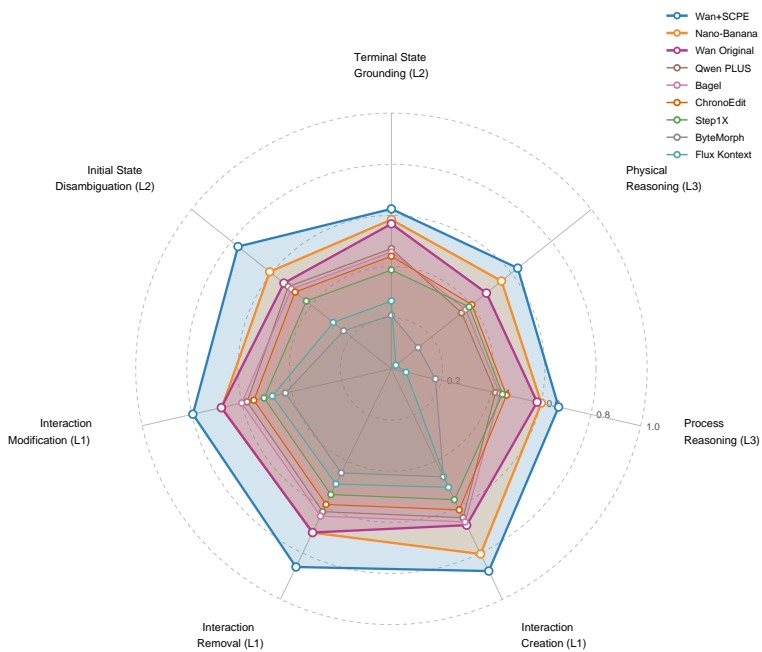

*Figure 12.* **Fine-grained interaction score across seven sub-dimensions of HOI-Edit.** For sub-dimensions in L1, we report **I**, for sub-dimensions in L2/3,we report **I+Q&A**

## D.2. More Visualization

Figure 13 further visualizes the performance advantages of our Wan+SCPE framework. By incorporating comparative results from TurboDiffusion and its SCPE-integrated variant, we demonstrate the robust generalization of SCPE across diverse video backbones. Notably, the integration of SCPE effectively mitigates "hallucination" (e.g., the motorcycle and jacket cases) and significantly enhances the precision of interaction editing, such as accurately connecting the alligator clip to the designated lemon. Furthermore, for long-range spatial repositioning (e.g., moving the person to a distant vanity table) where existing methods typically struggle with significant displacements, our approach exhibits superior spatial reachability.

Even in scenarios involving multiple concurrent interactions (e.g., the "pick up and stare at" actions in the Col. 5), our framework successfully achieves precise and coordinated editing.

Visualizations in Figure 14 confirm SCPE's superior handling of causal dependencies and physical effects. Compared to baselines, SCPE significantly enhances reasoning for intermediate interaction processes and physical properties. Specifically, it correctly identifies essential prerequisite actions—such as releasing a ball before standing (Col. 2) or unwrapping toast before eating (Col. 4)—whereas baselines produce structural artifacts like redundant limbs (e.g., Kontext, Step1) or contextually illogical edits (e.g., Nano Banana). Furthermore, SCPE ensures high fidelity by maintaining consistency between actions and their physical consequences (e.g., realistic sparks during metal polishing), while other models often fail to execute the edit or neglect corresponding physical phenomena.

## E. Limitations

Despite the strong performance of SCPE and the image-to-video (I2V)-based paradigm on complex human-object interaction (HOI) editing tasks, several limitations remain. First, existing video generation backbones still introduce relatively high inference time. Although the SCPE loop itself only adds lightweight agentic overhead and can be further accelerated by more efficient video generators, the proposed HOI editing paradigm still requires longer inference time compared with existing image editing methods.

Second, because SCPE relies on I2V backbones, it may inherit unintended camera motion, background drift, or non-target object motion from video generation models. This issue becomes more visible in scenes with multiple movable objects or complex backgrounds, and may impair the fidelity of background attributes. To quantify this limitation, we conduct experiments on HOI-Edit using the background evaluation protocol of LMM4Edit (Xu et al., 2025b). As shown in Table 8, Wan 2.2 I2V + SCPE achieves a background score of 0.5575, slightly improving over the base Wan 2.2 I2V model and surpassing the open-source static baseline Bagel. However, it still lags behind strong commercial static editors such as Nano Banana. This indicates that prompt-level constraints and Playbook-based correction can mitigate background instability to some extent, but cannot guarantee pixel-level background freezing. Although camera-control methods such as ReCamMaster (Bai et al., 2025) can be used to restrict camera motion and reduce background drift, they introduce additional processing time and may also cause potential visual distortion. Future work may further explore fine-grained camera control, background locking, and parameter-efficient adaptation strategies such as LoRA (Hu et al., 2022) to stabilize the generation process.

Finally, although HOI-Edit covers diverse interaction verbs and three cognitive levels, the current benchmark scale is still limited. Future work will expand the number of test samples, interaction categories, object types, and real-world scenarios to support more comprehensive evaluation of HOI editing models.

*Table 8.* **Quantitative Comparison on Background Preservation.** We report the background consistency scores proposed by (Xu et al., 2025b).

| Method | Background Score ↑ |
|---|---|
| ByteMorph (Chang et al., 2025) | 0.4648 |
| Bagel (Deng et al., 2025) | 0.5527 |
| Qwen-Image-Edit PLUS (Wu et al., 2025a) | **0.5769** |
| Nano Banana (Google, 2025) | 0.5760 |
| Wan 2.2 I2V (Original) | 0.5566 |
| Wan 2.2 I2V+SCPE | 0.5575 |
| Wan 2.2 I2V+SCPE+ ReCamMaster | 0.5806 |

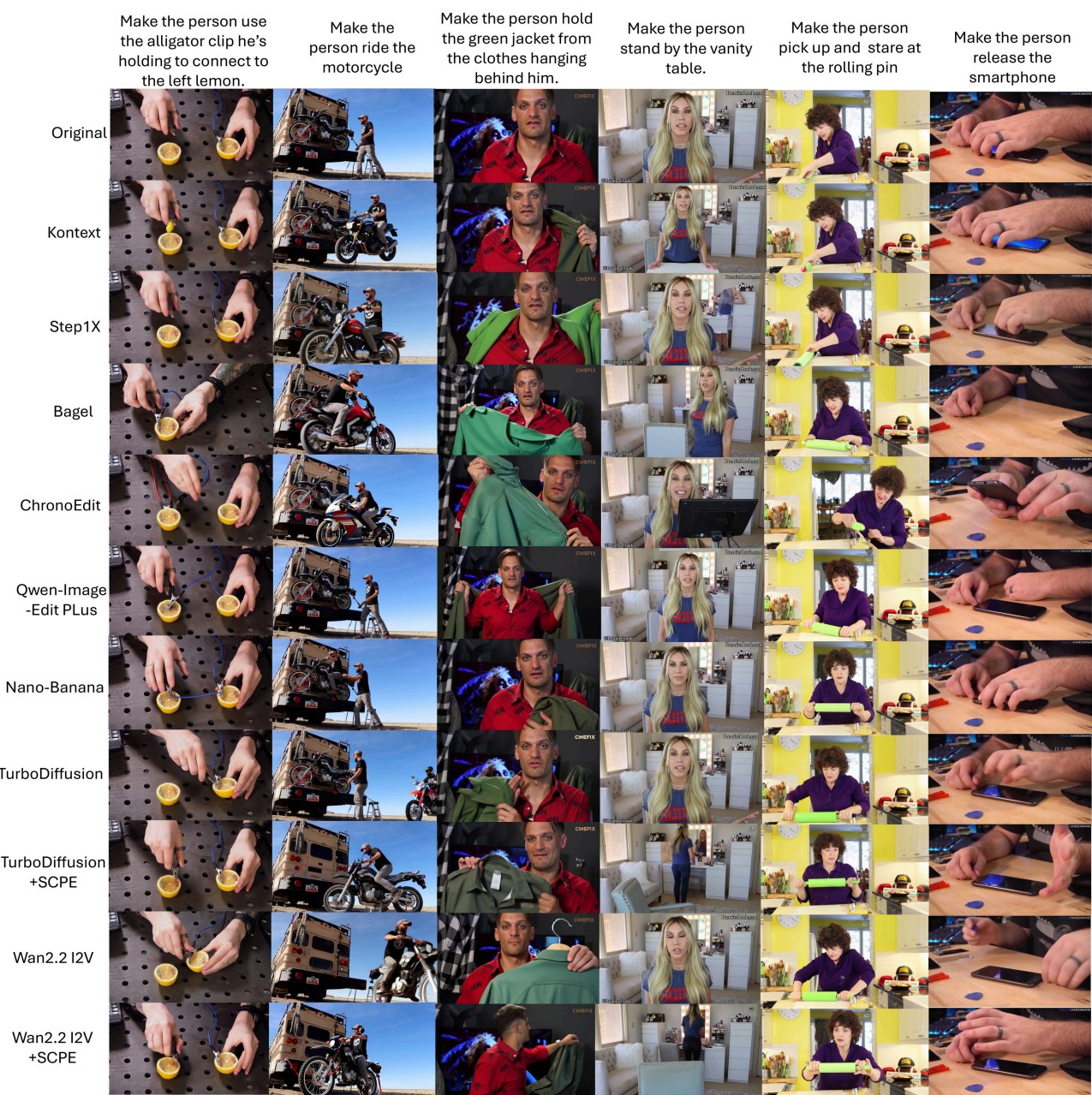

*Figure 13.* Qualitative comparison for L1 & L2.

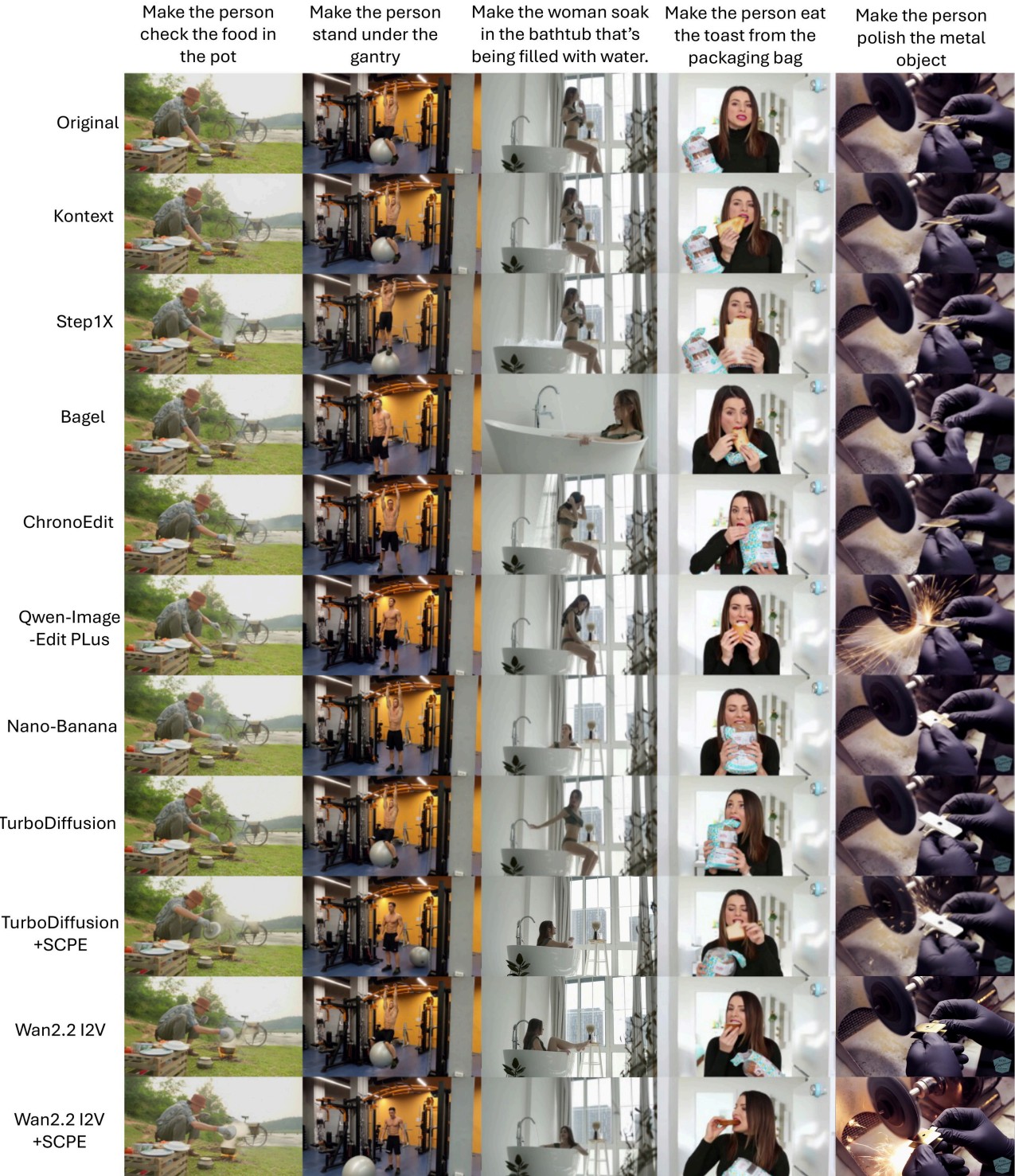

*Figure 14.* Additional visualizations on complex HOI scenarios involving phycial and causal chains.

