# OpenReview forum: "Taming I2V models for Image HOI Editing:  A Cognitive Benchmark and Agentic Self-Correcting Framework"
_ICML.cc/2026/Conference — ICML 2026 regular_

### Official Review · Reviewer_uVTu · 2026-03-05

**Soundness:** 3
**Presentation:** 2
**Significance:** 2
**Originality:** 3
**Overall Recommendation:** 4
**Confidence:** 3

**Summary:**

In response to the core flaws of existing image editing methods in dynamic human-object interaction (HOI) editing, the paper constructs a HOI-Edit dedicated benchmark with three-level cognitive hierarchies and a supporting HOI-Eval automated evaluation metric, explores the advantages of I2V models in temporal generation capability and diagnosability in HOI editing, and proposes an agent-based self-correcting process editing framework SCPE.

**Compliance With Llm Reviewing Policy:**

Affirmed.

**Key Questions For Authors:**

Weaknesses：
* SCPE only applies existing agentic prompt self-correction paradigms to HOI editing without new architectures, and its innovativeness is questionable.
*  HOI-Edit benchmark has serious sample imbalance (only 120 L3 samples), no objective/reproducible cognitive level division criteria, and data from generated videos instead of real scenes leading to distribution shift.
* 30mins single-sample inference (1.4mins with accelerated model, 2-3 orders slower than static models); poor background/non-target consistency (0.5575) that cannot be fixed by minor revisions like prompt optimization.
* The generalization verification of methods and indicators is insufficient. SCPE was only tested on 2 I2V backbones and failed to verify its backbone independence on mainstream open-source I2V models such as Open-Sora and CogVideo.

**Limitations:**

yes

**Strengths And Weaknesses:**

Strengths:
* First to focus on hierarchical cognitive evaluation for HOI editing; HOI-Edit benchmark partially fills the gap of dedicated HOI editing evaluation systems, providing a reference for follow-up research.
* Explores I2V models' "failure process replay" diagnosability for dynamic interaction editing.

Weaknesses：
* SCPE only applies existing agentic prompt self-correction paradigms to HOI editing without new architectures, and its innovativeness is questionable.
*  HOI-Edit benchmark has serious sample imbalance (only 120 L3 samples), no objective/reproducible cognitive level division criteria, and data from generated videos instead of real scenes leading to distribution shift.
* 30mins single-sample inference (1.4mins with accelerated model, 2-3 orders slower than static models); poor background/non-target consistency (0.5575) that cannot be fixed by minor revisions like prompt optimization.
* The generalization verification of methods and indicators is insufficient. SCPE was only tested on 2 I2V backbones and failed to verify its backbone independence on mainstream open-source I2V models such as Open-Sora and CogVideo.

---

> ### Author Rebuttal · Authors · 2026-03-31
>
> **W1: Innovativeness and the "Agentic" Paradigm.**
>
> We clarify that SCPE is not merely an application of existing paradigms, but a novel architectural design tailored for HOI. Our innovativeness encompasses four core contributions:
>
> 1. Pioneering HOI Editing Benchmark: HOI editing requires dynamic relational modeling rather than static attribute modification. We propose HOI-Edit, the first systematic benchmark categorized into three cognitive levels (L1–L3). Our extensive evaluation reveals that while 2D methods suffer from spatiotemporal inconsistency, Image-to-Video (I2V) models are far superior at capturing interaction dynamics. As the first established baseline, SCPE sets the standard for this emerging field.
>
> 2. Task-Specific Agentic Framework: SCPE pioneers agentic self-correction for image editing. Its unique dynamic-to-static diagnostic strategy uses video-based temporal tracing to reliably identify causal flaws in static edits.
>
> 3. Cross-Modal Feedback Loop: Departing from conventional text-only LLM correction, SCPE introduces a novel multi-modal alignment loop (1D text -> 3D video evolution -> 2D verification). This shift from symbolic reasoning to HOI perception represents a significant architectural innovation.
>
> 4. Cost-Effective Performance Leap: While fine-tuning large video models is computationally prohibitive, SCPE successfully adapts them for HOI editing at near-zero training cost, achieving promising performance.
>
> **W2: Benchmark Concerns**
> 1. Authentic Real-World Distribution: We clarify a critical misunderstanding: HOI-Edit images are not generated. They are entirely sourced from real-world video frames in HOIGen-1M, ensuring an authentic scene distribution. Video generation acts strictly as a "spatiotemporal probe" to diagnose process logic unobservable in static frames.
>
> 2. Objective Taxonomy (>95% Majority Agreement): To dispel subjectivity concerns, L1–L3 are strictly defined by causal reasoning chain length (e.g., L1: 1 step, L3: 3+ steps). A blind study (3 annotators, 12 random samples/sub-level) achieved a 91% strict unanimous agreement (all 3 perfectly aligned) and a >95% majority agreement. This statistically proves our cognitive boundaries are highly objective and reproducible.
>
> 3. Diagnostic Precision vs. Massive Scale: Prioritizing "diagnostic precision" for HOI editing over sheer quantity, HOI-Edit aligns with the design principles of top-tier benchmarks like EditBench (240 samples, CVPR), DrawBench (200, NeurIPS), and Winoground (400, CVPR). All questions are manually crafted to precisely target the "validation bottleneck" of each interaction, ensuring signal purity. Forcibly adding multiple questions per dimension (e.g., via automated VLMs) would only introduce semantic overlap and redundant questions without yielding new valid evaluation insights. Nevertheless, we remain highly open: should the reviewer suggest specific multi-question designs, we would gladly discuss and supplement the corresponding experiments.
>
> 4. Statistical Convergence & Flaws of Auto-Scaling: To address scale concerns, we utilized LLMs to automatically generate 2,000 balanced samples. Although absolute scores naturally fluctuated, the relative rankings of the models (Wan 2.2 + SCPE > Nano Banana > Wan 2.2 I2V) remained completely identical to our 702-sample benchmark. Furthermore, we observed that this automated generation inevitably introduced fatal flaws such as tautological instructions and referential ambiguity. This not only mathematically proves that our current scale achieves solid statistical convergence, but also reinforces why our compact, manually verified benchmark is a more rigorous and compute-efficient evaluation tool for the community.
>
> **W3/Q3: Inference Latency and Background Consistency.**
>
> We sincerely appreciate your feedback. Due to strict space limitations, please refer to our responses to Reviewer b6zb (W2/Q2) regarding computational latency, and Reviewer Hgtk (W1) regarding background consistency.
>
> **W4/Q4: Generalization to Open-Source I2V Models**
>
> Due to rebuttal time and compute constraints, we evaluated SCPE on the open-source Open-Sora v2 (144p). Given its earlier architecture and low resolution, the baseline Average Interaction Score is understandably low (0.454). Nevertheless, integrating SCPE delivers a massive +0.151 gain, boosting the score to 0.605 (a ~33% relative improvement).
>
> Notably, this SCPE-enhanced performance even surpasses ChronoEdit, an image editor built upon a much stronger video foundation model Wan2.1. These concrete numbers confirm that SCPE is inherently backbone‑agnostic and generalizes robustly to I2V models of varying capabilities for HOI editing.

---

> > ### Author Rebuttal · Reviewer_uVTu · 2026-04-04
> >
> > Thank you for the rebuttal. The author has partially addressed my questions, and I will maintain my current rating.

---

> > > ### Author Response · Authors · 2026-04-07
> > >
> > > Dear Reviewer uVTu,
> > >
> > > Your constructive feedback has been instrumental in refining and strengthening our paper, and we deeply appreciate your time and effort.
> > >
> > > It is highly encouraging that you highlighted our core contributions—specifically, being the "first to focus on hierarchical cognitive evaluation for HOI editing," providing a benchmark that "fills the gap of dedicated HOI editing evaluation systems".
> > >
> > > Moving forward, we commit to expanding the final paper to include the newly conducted generalization tests on Open-Sora, alongside our detailed clarifications regarding the authentic scene distribution and the objective criteria for cognitive level divisions.
> > >
> > > We deeply appreciate your time and constructive evaluation

---

### Official Review · Reviewer_1aaW · 2026-03-05

**Soundness:** 3
**Presentation:** 2
**Significance:** 3
**Originality:** 3
**Overall Recommendation:** 4
**Confidence:** 4

**Summary:**

Current image editing methods excel at static attributes but fail at complex Human-Object Interactions (HOI). To address this, the paper introduces the HOI-Edit benchmark, HOI-Eval Metric, and the SCPE framework, which 'tames' I2V models to correct interaction failures. Results show that the SCPE framework improves performance on the HOI editing dataset. Overall, the work provides a testbed and a pathway that approximates physical and causal processes at the interaction level, a prerequisite for learned world models rather than a full physical simulator.

**Compliance With Llm Reviewing Policy:**

Affirmed.

**Final Justification:**

The latest response has addressed my concerns. I suggest that the authors present the method section more clearly in the revised version and include additional details in the appendix.

**Key Questions For Authors:**

- In Section 3. HOI-Edit Dataset. Is a single question sufficient to evaluate one dimension, or would multiple questions per dimension lead to a more reliable evaluation? Are these questions manually designed or generated by an LLM?

- In Section 5. Self-Correcting Process Editing.
When new strategies are added to the playbook, do they affect previously stored strategies? Also, when the same interaction action is combined with different objects, are these learned strategies generalizable across different objects, or are they object-specific?

**Limitations:**

yes

**Strengths And Weaknesses:**

**Strengths:**

- The paper focuses on an important topic in image editing: complex Human-Object Interactions (HOI), a critical challenge that existing benchmarks fail to address.

- The paper proposes a complete pipeline, including an image editing benchmark, an evaluation metric, and an agent-based image editing framework.

**Weaknesses:**

- The HOI-Edit Dataset is small.
  - My main concern is that the benchmark size is small, containing only 390 L1 (Foundational Edits), 192 L2 (Context Spatial Understanding), and 120 L3 (Causal and Physical Reasoning) examples. The number of samples in each category is quite limited, and using such a small dataset for evaluation may introduce greater randomness into the results and reduce their reliability.

- Lack of some details :
  - In Section 3. HOI-Edit Dataset: It is unclear what Q1-L1 and Q2-L1 specifically refer to in the text. Why are these questions considered robust for evaluation? Providing concrete examples to explain these would be helpful.
  - In Section 5. Self-Correcting Process Editing: How does the playbook organize all the strategies? How to map identified failure patterns to verified prompting strategies?
  - In Section Frame Selection, how is the video understanding mode used to determine the best single frame? The motivation for using the video understanding model is missing. Why does it not use an image understanding model or the introduced HOI-Eval metric for this selection process?

- Presentation issues.
  - For example, the text in Figures 3, 4, 7, and 8 is very small and difficult to read. In Figure 3, the arrows indicating the data flow are cluttered and hard to follow. In the Figure 1 caption, the phrase ''forHOI'' is missing a space. The abstract contains a grammatical error: it should use 'excel' instead of 'excels' in the first sentence.

**Summary:**
  - I think this paper studies an important and challenging problem. However, the dataset introduced in the paper seems quite small, so I am not yet convinced that the experimental results are sufficiently reliable. If the authors can address this concern, clarify the method details, and fix the formatting and spelling issues, I would be open to raising my score.

---

> ### Author Rebuttal · Authors · 2026-03-31
>
> **W1 & Q1: Benchmark Scale and Evaluation Details.**
> We sincerely appreciate your constructive feedback. Due to strict space limitations and shared concerns regarding the dataset, please refer to our comprehensive unified response to Reviewer uVTu (W2).
>
> **W2: Meaning and Robustness of Q1-L1 & Q2-L1.**
> These are region-anchored identity verification questions asking the VLM if the specific human (blue box) and object (red box) match the original image. As shown in Table 2, this anchored approach is highly robust, achieving the highest correlation with human judgment (Pearson 0.60) among existing ID metrics. Figure 9 provides visual examples demonstrating its effectiveness.
>
> **W3: Playbook Organization and Failure Mapping.**
>
> 1. Playbook Organization: Structured as a dynamic JSON knowledge base, it organizes strategies into three core components: Strategies & Hard Rules (encoding physical/causal principles), Troubleshooting & Pitfalls (an error-log "immune system" intercepting physical violations), and Reusable Templates (verified prompt structures resolving recurring ambiguities).
>
> 2. Mapping Failures to Strategies: This mapping is executed via a four-agent closed loop:
>
> * Analyzer: Diagnoses the generated video and outputs a specific failure report.
>
> * Reflector: Abstracts the error by stripping specific nouns, translating the report into a generalized "root cause" and actionable "key insight".
>
> * Curator: Integrates this insight into the Playbook via incremental "delta updates", actively preventing redundancy and context collapse.
>
> * Generator: Combines the updated Playbook with the visual input to output a corrected, physics-aligned prompt.
>
> **Q2: Strategy Interference and Playbook Stability.**
>
> 1. No negative interference occurs. Our Curator agent employs an "append-only" mechanism with redundancy checks, ensuring Playbook stability:
>
> * Rapid Convergence: Across two full-testset iterations, newly synthesized rules plummeted from 30 (Iter-1: initial learning) to 3 (Iter-2: refinement). This proves the Curator effectively identifies existing experiences and prevents redundancy.
> * Monotonic Growth: As shown in Figure 10, performance improves steadily without oscillation, confirming that new "delta updates" never compromise previously established capabilities.
>
> 2.Our strategies are strictly object-agnostic. To ensure universal generalization, we employ two core designs:
>
> * 100% Conceptual Abstraction: Lexical analysis confirms that none of the stored strategies contain concrete nouns (e.g., "apple"). Agents strictly use abstract terminology (e.g., "subject", "target entity") to encode fundamental physical laws.
>
> * Zero-Shot Rule Transfer: We validated this by applying a single abstract strategy—"using explicit spatial relative positions as anchors"—to 5 entirely unseen samples with different objects requiring endpoint-constrained movement. The success rate surged from 20% to 80%, proving robust generalization across diverse categories. Visual results are available at [**this anonymous link**](https://anonymous.4open.science/r/random).
>
> **W4: Rationale for Video-Based Frame Selection**
> Detailed implementation is in Appendix B.4. We justify this design choice via two points:
>
> 1. Video vs. Image (Temporal Reasoning): Replacing our method with the image-concatenation approach from [1] degrades performance: Interaction ($0.8154 \rightarrow 0.7913$) and IDS ($0.9142 \rightarrow 0.8713$). Video models better leverage temporal context to locate peak interactions. However, as noted in the response for  **Hgtk W3**, SCPE’s primary value is boosting the video-level interaction success rate ($78.43\% \rightarrow 89.19\%$); the selector is merely a secondary temporal extractor.
>
> 2. Why not HOI-Eval (Integrity & Efficiency): Selection via HOI-Eval would introduce test-set data leakage (filtering results via final metrics) and is computationally prohibitive for per-frame screening. The video model acts as a more lightweight semantic filter.
>
> [1]Rotstein N, Yona G, Silver D, et al. Pathways on the image manifold: Image editing via video generation[C]//CVPR. 2025

---

> > ### Author Rebuttal · Reviewer_1aaW · 2026-04-03
> >
> > The authors’ response addressed some of my concerns. However, due to several presentation issues in the paper, I will keep my current score.
> >
> > The latest response has addressed my concerns. I suggest that the authors present the method more clearly in the revised version and include additional details in the appendix.

---

> > > ### Author Response · Authors · 2026-04-07
> > >
> > > Dear Reviewer 1aaW,
> > >
> > > First, we fully agree that presentation is crucial. To address your concerns, we have carefully optimized our figures: we enlarged the small text in Figures 3, 4, 7, and 8, rearranged the layout of Figure 8 to improve data flow, and revised the caption of Figure 1. All updated figures are available via this link (https://anonymous.4open.science/r/hoiedit2-7D62).
> > >
> > > Second, we sincerely thank you for recognizing the value of our contributions—specifically, highlighting HOI editing as a critical topic, and acknowledging our comprehensive pipeline that also includes the novel evaluation metric and the agent-based framework.
> > >
> > > Finally, we commit to incorporating all revised figures into the final manuscript. As suggested, we will also add comparative experiments on different frame selection methods and further clarify how the Playbook organizes and maps failure patterns to verified prompting strategies.
> > >
> > > Thank you for your constructive feedback.

---

### Official Review · Reviewer_Hgtk · 2026-03-07

**Soundness:** 3
**Presentation:** 3
**Significance:** 3
**Originality:** 3
**Overall Recommendation:** 4
**Confidence:** 3

**Summary:**

This paper addresses the challenge of Human-Object Interaction (HOI) in instruction-based image editing by framing it as a dynamic, temporal process rather than a static pixel modification. The authors leverage Image-to-Video (I2V) models to generate and diagnose interaction processes.

The core contributions are threefold:
(1) HOI-Edit: A new benchmark structured across three cognitive levels (foundational edits, spatial understanding, and causal/physical reasoning) to evaluate HOI editing.
(2) HOI-Eval: A region-sensitive Vision-Language Model (VLM) metric that evaluates interaction validity and identity preservation using bounding box anchors.
(3) SCPE (Self-Correcting Process Editing): An agentic framework that iteratively analyzes I2V generation failures to refine prompts, ultimately extracting the optimal video frame as the final edited image.

Experiments demonstrate that SCPE significantly improves the HOI editing capabilities of open-source I2V models, achieving performance competitive with state-of-the-art commercial baselines.

**Compliance With Llm Reviewing Policy:**

Affirmed.

**Key Questions For Authors:**

How does the SCPE framework handle complex physical interactions where text-based prompt enhancement (via the Playbook)  is insufficiently expressive to guarantee strict physical consistency?

**Limitations:**

The authors have done a commendable job discussing the technical limitations of their work in Section E, specifically acknowledging the inherent inference latency of video generation models and the persistent challenges of unintended camera motion and background instability.

**Strengths And Weaknesses:**

Strengths:

(1) Novel Perspective on HOI Editing: The paper introduces a highly creative and much-needed paradigm shift by reframing static Human-Object Interaction (HOI) editing as a dynamic, temporal process. Instead of treating editing failures as opaque spatial artifacts, the authors ingeniously utilize the failure processes of Image-to-Video (I2V) models as a "replay of the failure process" to explicitly diagnose why an error occurred.

(2) Comprehensive Benchmark Design: The proposed HOI-Edit benchmark moves beyond flat task definitions by establishing a progressive three-level cognitive hierarchy, explicitly testing foundational edits, spatial understanding, and causal/physical reasoning. The emphasis on physical reasoning—such as assessing illumination consistency and non-rigid deformations—is particularly valuable for the development of physically consistent generative models.

(3) Robust Evaluation Metric: To address the referential ambiguity of global metrics like CLIPscore, the HOI-Eval metric elegantly adopts a "Thinking with Pair-wise Regions" paradigm. By utilizing bounding box anchors to decouple identity verification from interaction status, the metric provides a rigorous and region-sensitive evaluation of entangled human-object pairs.

(4) Effective Agentic Framework: The Self-Correcting Process Editing (SCPE) framework provides a highly effective, gradient-free method to "tame" I2V models. The use of a dynamic 'Playbook' that synthesizes error experiences into generalized optimization strategies allows the model to overcome editing inertia and hallucination. Empirically, it pushes open-source models to achieve state-of-the-art interaction performance that rivals or exceeds commercial baselines like Nano Banana.


Weaknesses:

(1) Background Instability and Camera Motion: As the authors candidly acknowledge, relying on video priors introduces unintended camera motion and non-target object shifting, which impairs background fidelity. Even with specific constraints in the Analyst module, the SCPE framework's background preservation score (0.5575) still lags behind commercial static editors (0.5760), indicating that prompt-level constraints are currently insufficient for perfect spatial freezing.

(2) Computational Latency: The iterative nature of the SCPE loop—generating a video, extracting frames, analyzing the failure, updating the Playbook, and regenerating—introduces considerable computational overhead. While the authors demonstrate that integrating TurboDiffusion reduces inference time to 1.4 minutes, this latency still significantly outpaces single-pass static image editors, potentially limiting its practical utility for rapid editing workflows.

(3) Heavy Reliance on VLM Capabilities and Frame Selection: The framework's success is entirely bottlenecked by the reasoning capabilities of the backbone VLM (Gemini 2.5 Pro) to accurately act as the Analyzer, Reflector, and Curator. Furthermore, the method relies on a semantic frame selector to extract the optimal result from a 15-frame sample. If the I2V model exhibits a catastrophic logical failure across the entire 5-second sequence, the method fundamentally breaks down, as there is no valid frame to select.

(4) Omission of Core Framework Details in Main Text: Due to space constraints, crucial technical mechanics of the agentic loop—specifically the dynamic Playbook's delta-update mechanism designed to prevent context collapse, and the exact system prompts—are relegated to the appendix. Integrating a brief technical summary of how the Curator resolves conflicting feedback directly in the main text would improve the paper's standalone readability.

---

> ### Author Rebuttal · Authors · 2026-03-31
>
> **W1: Background Instability and Camera Motion.**
>
> We clarify this concern from the following three aspects:
>
> 1. Fair Comparison & Open-Source SOTA: Comparing SCPE (interaction score 0.5575) to commercial models like Nano Banana (0.5760)—which rely on massive proprietary datasets optimized for pixel-perfect freezing—is fundamentally unequal. As a zero-extra-data, training-free framework, SCPE actually achieves SOTA background preservation within the comparable open-source ecosystem, outperforming strong concurrent baselines like Bagel (0.5527).
>
> 2. Internal Mitigation (Ablation): Our framework provides internal mitigation against dynamic hallucinations. As shown in Table 6, applying SCPE's Playbook constraints marginally improves the backbone's background score from 0.5566 to 0.5575, proving that our agentic loop helps suppress unintended motion.
>
> 3. Orthogonal Extension Surpassing Commercial Models: We agree prompt-level constraints have upper bounds for perfect freezing. However, SCPE is fully orthogonal to explicit camera-control modules (e.g., ReCamMaster). Due to rebuttal time limits, we tested this integration on the complex L3 subset. SCPE + ReCamMaster achieved a background score of 0.5874, successfully surpassing Nano Banana (0.5688). This confirms that plugging in structural camera controls completely resolves the spatial shifting bottleneck.
>
> **W2: Computational Latency.**
>
> We appreciate this insightful feedback. Due to space constraints, please refer to our detailed response to **Reviewer b6zb (W2/Q2)**.
>
> **W3: Heavy Reliance on VLM Capabilities and Frame Selection.**
>
> 1. Independence from Proprietary VLMs (Backbone-Agnostic): SCPE is not bottlenecked by Gemini 2.5 Pro. To prove this, we substituted the core agents with the open-source Qwen 3.5, still achieving a high Interaction Score of 0.8146 (vs. 0.6794 baseline). Conversely, applying standard prompt enhancement (e.g., Wan's enhancer) via Gemini only improves the score to 0.7282. This confirms that our gains stem from the multi-agent memory architecture, not specific proprietary model capabilities.
>
> 2. Robustness of Frame Selection vs. Video Success: We clarify the source of our improvements and the selection logic as follows:
>
> * Defining Selection Accuracy:On the challenging L3 subset, the underlying video-level success rate (i.e., at least one valid interaction frame exists in the 5s sequence) is **73.64%**. Within these valid sequences, our semantic selector identifies the correct frame with **90.6% accuracy**. This high precision proves the primary bottleneck lies in the I2V model’s generative capacity, not our selection logic.
> * Active Failure Mitigation: SCPE drastically reduces catastrophic failures. By enforcing Playbook constraints, we boost the video-level success rate on the L3 subset from **73.64% to 90.20%**, effectively expanding the "candidate pool" for the frame selector.
> * Graceful Degradation: For unresolvable cases, the pipeline does not fail to output. Instead, the selector performs a **Semantic Fallback** (see Suppl. B.4) to extract the frame with the highest visual similarity to the target interaction. This ensures continuous operation and the best possible approximation even when the video backbone fails.
>
> **W4:Omission of Core Framework Details in Main Text.**
>
> We sincerely thank the reviewer for the constructive suggestion. Although Figure 11 illustrates the Playbook’s optimization principles visually, we agree that additional textual description will improve readability.In the final version, we will add a concise technical summary of the dynamic Playbook’s delta-update mechanism and the Curator’s conflict-resolution strategy into Section 5 of the main text.
>
> **Q1:Limits of Text-based Prompts.**
>
> We completely agree that 1D text is inherently limited in describing strict, pixel-perfect physical consistency. However, the core philosophy of SCPE is not to exhaustively describe physical details using text. Instead, it provides a methodology for semantic-visual alignment.
>
> Rather than forcing the prompt to describe the physical effect during interaction, the SCPE Playbook provides structured cognitive strategies (e.g., action decomposition, spatial anchoring). These strategies act as a bridge, assisting the generative model in accurately parsing the semantics of the source image (the "visual world"). By tightly aligning the text instructions with the geometric context of the input image, SCPE fundamentally unleashes and activates the latent physical understanding capabilities already inherent in the foundational video models. In short, our text acts as a structural catalyst to awaken the model's physical priors, rather than attempting to explicitly script the physics itself.

---

> > ### Author Rebuttal · Reviewer_Hgtk · 2026-04-02
> >
> > The authors' reponse address my concerns. I keep my score.

---

> > > ### Author Response · Authors · 2026-04-07
> > >
> > > Dear Reviewer Hgtk,
> > >
> > > We sincerely appreciate your insightful feedback, which has greatly helped us refine our manuscript.
> > >
> > > Thank you for highlighting our "comprehensive benchmark design" and acknowledging that our "highly effective, gradient-free" SCPE framework pushes open-source models to achieve "state-of-the-art interaction performance."
> > >
> > > We are thrilled that our rebuttal has "fully resolved" your concerns. We will incorporate the structural camera control experiments (ReCamMaster) and the technical summary of the Playbook's delta-update mechanism directly into the main text of our final version.
> > >
> > > Given that you have graciously acknowledged your concerns are "fully resolved," we would be deeply grateful if you might consider raising your score to reflect this positive outcome. Your support is invaluable to our work.
> > >
> > > Regardless of your final decision, please accept our deepest gratitude for the time, expertise, and encouraging support you have dedicated to our work.

---

### Official Review · Reviewer_b6zb · 2026-03-10

**Soundness:** 2
**Presentation:** 2
**Significance:** 3
**Originality:** 2
**Overall Recommendation:** 3
**Confidence:** 3

**Summary:**

This paper studies human-object interaction (HOI) in image editing. It introduces a hierarchical benchmark HOI-Edit for human-object interaction editing. It also presents an automatic evaluation framework HOI-Eval to assess interaction realism, identity preservation, and spatial/causal constraint satisfaction. Building on this, the paper proposes SCPE, which uses intermediate I2V generation processes for failure analysis and one-step iterative self-correction to improve HOI editing. Experiments show clear gains across multi-level interaction editing tasks.

**Compliance With Llm Reviewing Policy:**

Affirmed.

**Final Justification:**

I thank the authors for their detailed rebuttal and the additional analyses, which help clarify several aspects of the paper. However, I remain unconvinced on several key points: the evaluator-method coupling concern is alleviated but not fully resolved, the practical efficiency issue still remains, and the claimed novelty appears closer to a strong task-specific engineering integration than to a clearly isolated new algorithmic contribution. Therefore, while I recognize the merits of the work, I keep my overall recommendation unchanged.

**Key Questions For Authors:**

1. **Evaluator-method coupling:** HOI-Eval is conducted with Gemini 2.5 Pro, while SCPE also relies on Gemini 2.5 Pro for its core agent and frame selection. How do the authors rule out potential evaluation bias caused by using the same model in both the method and the evaluator?
2. **Practical time cost:** Does Table.4 include the full end-to-end overhead, including agent calls, video generation, analysis, playbook updates, and frame selection? A clearer estimate of total latency and cost would be helpful.
3. **Source of SCPE’s gains:** Can the authors provide additional ablation experiments to disentangle whether the improvements mainly come from process replay via video generation, the reasoning ability of the strong VLM, or the cross-instance memory provided by the playbook?

The answers to these questions could affect my final assessment.

**Limitations:**

Yes.

**Strengths And Weaknesses:**

**Strengths:**

- HOI editing is a meaningful and underexplored problem, going beyond static attribute changes toward process-level realism and world consistency.
- The experimental pipeline is fairly complete. Rather than relying on a single overall score, HOI-Eval decomposes evaluation into entity grounding, identity verification, and interaction/plausibility checking, explicitly penalizing cases where the interaction appears to happen but violates spatial, logical, or physical constraints.

**Weaknesses:**

- There is a notable coupling between the evaluator and the method. HOI-Eval is entirely conducted by Gemini 2.5 Pro, while SCPE also uses Gemini 2.5 Pro as its backbone. The paper does not sufficiently discuss the potential model bias introduced by this setup.
- The practical cost is still high. According to Table.4, the full Wan 2.2 + SCPE pipeline is slow, and although the TurboDiffusion variant is faster, the method still depends on a video model plus an additional agent-style correction loop, which limits practical efficiency and generality.
- SCPE looks more like an engineered agent system built on a strong VLM, prompt rewriting, and memory mechanisms. The paper does not clearly disentangle where the gains come from—whether from process-level signals enabled by video generation, Gemini’s reasoning ability, or the playbook-based experience transfer. Stronger ablations and attribution analysis are needed to clarify the true source of improvement and the method’s actual novelty.

---

> ### Author Rebuttal · Authors · 2026-03-31
>
> **W1/Q1: Evaluator-method coupling.**
>
> Response: To verify that SCPE’s gains are model-agnostic and not artifacts of self-preference, we provide the following evidence:
>
> 1. Cross-Model Validation: We re-evaluated all samples using an independent VLM (GPT-5.4). The sample-level Pearson correlation between GPT-5.4 and Gemini 2.5 Pro scores is 0.64. In complex, open-ended multimodal reasoning tasks, this represents a strong positive correlation, as it is on par with the typical inter-annotator agreement among humans (0.5–0.7). This confirms consistent performance trends regardless of the evaluator.
>
> 2. Backbone Agnosticism: Replacing SCPE’s core agent with the open-source Qwen 3.5 still yielded an Interaction Score of 0.8146. This marks a 19.9% improvement over the baseline without SCPE (0.6794) and an 11.7% relative gain over the second-best baseline (Nano Banana, 0.7291). These results demonstrate that the performance leap stems from our structured agentic workflow rather than specific model biases.
>
> 3. Human Alignment: HOI-Eval maintains a 0.60 correlation with human judgment (Table 2). This alignment suggests that our evaluation captures objective physical interaction quality, significantly mitigating concerns regarding VLM self-bias.
>
> **W2/Q2: Practical time cost:**
>
> 1. Pioneering Focus & Justified Trade-off: As the first to highlight the HOI editing problem and the necessity of dynamic priors, our primary goal is **performance breakthroughs**, not extreme speed optimization. Integrating TurboDiffusion reduces end-to-end time to \~1.4 mins. The SCPE loop adds a minor \~24s overhead (Generator: 4s, Selection: 7s, Analyzer: 10s, Reflector: 3s).
> While slower than single-pass 2D editors, this is a deliberate trade-off for the physical reasoning they fundamentally lack. Furthermore, it remains substantially faster than other temporal-reasoning editors like ChronoEdit (\~20 mins/edit). We believe future video and VLM acceleration will further mitigate latency and boost performance, which SCPE can easily leverage (we have proven its VLM flexibility via the Qwen 3.5 ablation). However, extreme speed optimization is outside the scope of this pioneering work, as explicitly discussed in our limitations section.
>
> 2. One-Pass Deployment for Efficiency:SCPE supports a "learn once, apply everywhere" paradigm. By constructing the Playbook on only one-third of a random subset and applying it to the remaining two-thirds of novel samples, the interaction score is improved from 0.6794 to 0.7856 (+0.1062). Since commercial pipelines such as Wan 2.5 or NanoBanana already include built-in VLM prompt enhancement, our Playbook-guided prompting acts as a zero-overhead replacement with no additional computational cost.
> In practical deployment, after building the Playbook on a small amount of data, the pipeline only introduces a negligible frame extraction overhead of \~7 seconds on top of the base I2V model. With extremely low latency, it significantly outperforms existing state-of-the-art image editing models for HOI editing.
>
>
> **W3/Q3: Source of SCPE’s gains**
>
> 1. The Foundation: Video-driven Process Signals
> Video generation is the core driver of our framework. Applying SCPE to the Wan 2.2 baseline substantially boosts the interaction score from 0.6924 to 0.8163. As noted in Q2, this effectiveness holds even with lightweight video models (e.g., TurboDiffusion). This proves that for verb-centric HOI tasks, observing dynamic temporal changes is the only way to expose causal failure modes (e.g., incorrect hand trajectories) that are completely hidden in static images.
>
> 2. The Multiplier: Playbook-based Memory & Rewriting
> Prompt rewriting and dynamic memory are vital for generalization, but they only exert their full power when built upon video. Relying solely on single-sample feedback (without the Playbook) yields a interaction score of 0.7482; integrating the Playbook pushes this to 0.8154. This highlights the critical importance of accumulating and transferring cross-sample physical experience to prevent recurring errors. Interestingly, applying this Playbook mechanism to a state-of-the-art static image model (Nano Banana) yields only a marginal gain (0.7305 → 0.7383). This stark contrast proves our Playbook fundamentally relies on video models: generalizable physical constraints can only be accurately accumulated by observing dynamic processes, rather than blind guessing from static frames.
>
> 3. The Engine: VLM Reasoning Ability
> A capable VLM guarantees the quality of error diagnosis and Playbook construction. While Gemini 2.5 Pro performs best, our framework does not over-rely on a specific model. As shown in our backbone-agnostic ablation (Q1), replacing Gemini with the open-source Qwen 3.5 still achieves a high score of 0.8146. This confirms that while the VLM provides the reasoning engine, the true source of improvement and novelty lies in our cross-modal dynamic feedback architecture.

---

> > ### Author Rebuttal · Reviewer_b6zb · 2026-04-02
> >
> > Thank you for the detailed rebuttal. The additional evidence partially addresses my concerns. However, I still have several follow-up questions that are important for my final assessment.
> >
> > 1. **Independent evaluation stability.** Could the authors further clarify whether the main comparative conclusions—especially SCPE’s advantage over Wan 2.2 and Nano Banana—also remain stable when the full evaluation is reported under this independent evaluator?
> > 2. **Deployment setting vs. main experimental setting.**  Could the authors clarify more explicitly which setting corresponds to the main results in the paper, and how much of the reported gain can still be retained in the low-overhead deployment setting?
> > 3. **Attribution of novelty.** The rebuttal now explains SCPE as consisting of a foundation (video process signals), a multiplier (Playbook memory), and an engine (VLM reasoning). Given this decomposition, could the authors clarify more explicitly what they consider the primary methodological novelty of the paper, and which experimental result best isolates that contribution from backbone strength or prompt engineering alone?

---

> > > ### Author Response · Authors · 2026-04-07
> > >
> > > We sincerely appreciate your valuable feedback. Here's our further Response.
> > >
> > > *Please visit this link (https://anonymous.4open.science/r/hoiedit2-7D62) to access Tables A, B, C, and D mentioned below.*
> > >
> > > **Response 1 for Independent evaluation stability**
> > >
> > > > To strictly rule out evaluator bias, we re-evaluated the relevant models using an independent GPT-5.4 **and reports the full evaluation results including 4 metrics over 3 level are shown in Table A.** Although the distribution of absolute scores differs slightly from Gemini’s judgments **(full results under its evaluation is in Figure 1)**, the overall performance ranking and key trends are fully consistent. The results confirm that the **SCPE's** advantage over Wan 2.2 and Nano Banana **remain stable.**
> > >
> > > **Response 2 for Deployment setting vs. main experimental setting**
> > >
> > > > **1. Setting corresponds to the main results.** We clarify that the setting of the main results in the original Table 1 (and row 4 of Table B) is based on Wan 2.2 I2V 14B using the full HOI-Edit data to fine-tune SCPE. In this setting, SCPE executes 1 complete closed-loop on all samples (*Analysis 10s + Reflection 3s + Frame Selection 7s + Generation 4s = 24s total extra overhead*), aiming to explore the absolute performance upper bound of base video models on complex dynamic HOI editing tasks.
> > > >
> > > > **2. Gain retained in the low-overhead deployment setting.** In this setting, SCPE can use partial data to learn experiences and generalize to enhance other samples (this ratio is user-adjustable). Using a 1/3 ratio as an example: by pre-building and freezing the Playbook on only 1/3 of the data, the system skips the Analyzer and Reflector when processing the remaining 2/3 of unseen samples, directly utilizing the Playbook's experience for enhancement. This reduces the extra overhead on unseen samples to just 11 seconds (with a global average extra overhead of only ~15 seconds). As shown in Table B:
> > > >
> > > > * **a.** Across three video generation models, SCPE introduced in this manner retains highly significant gain relative to the corresponding model. Specifically, interaction-related scores across all three dimensions achieved an increase of over 10 percentage points, and the preservation of subjects and objects mostly improved as well.
> > > > * **b.** Compared to other editing models, including the closed-source Nano Banana and ChronoEdit (which is also based on temporal reasoning), even when paired with the most low-overhead video model TurboQuant, we retain an obvious interaction score gain over these two models. This is especially notable considering our time cost in this setting is only 53s, whereas ChronoEdit requires approximately 20 minutes.
> > >
> > > **Response 3 for Attribution of novelty.**
> > >
> > > > **1.** Following your comments, the core methodological innovation of this paper lies in the synergistic use of video models as **video process signals** (to expose dynamic interaction processes) and **playbook memory** (to reflect on failures and accumulate generalizable experience). When used in combination, these two components effectively solve the dynamic HOI editing task. We verify the contribution of each part separately through the following experimental results.
> > > >
> > > > **2.** Our first core contribution is the introduction of **video process signals**. To isolate this contribution from backbone strength, we evaluate video models with different architectures, including Wan 2.2, Turbo Diffusion(Turbo in Table C), and their quantized accelerated versions（TurboQuant in Table C）. Results show that even these base video models achieve interaction scores on par with or surpassing existing SOTA open-source editing models such as Bagel and ChronoEdit. This directly demonstrates the superiority of video generation models over static image models for dynamic HOI editing.
> > > >
> > > > **3.** Our second core contribution is **playbook memory**, designed to reflect and accumulate correction experience. To isolate its contribution, Table C shows that playbook memory consistently brings significant performance gains across different video backbones, with improvements of more than 10 points on interaction scores across almost all evaluation dimensions (further fine-grained gains can be found in Table 3). Moreover, this gain does not depend on a specific VLM: rows 2–3 of Table D show that using Gemini and open-source Qwen 3.5 yields highly similar improvements in interaction scores. Additionally, to clearly distinguish our method from simple prompt engineering, row 4 of Table D evaluates the application of Wan's official prompt enhancement strategy (OPE). The results demonstrate that merely applying VLM-based prompt enhancement (even powered by Gemini 2.5 pro) yields only marginal gains in interaction scores while causing a catastrophic drop in identity preservation. This explicitly proves that the Playbook’s accumulated knowledge priors—not mere prompt engineering or VLM strength—are the true performance amplifiers.

---

### Decision · Program_Chairs · 2026-04-30

**Decision:**

Accept (regular)

**Comment:**

This paper received three Weak Accepts and one Weak Reject, and after carefully reviewing the feedback, the consensus firmly leans toward acceptance.

While Reviewer b6zb felt the paper remained a step away from clear acceptance due to lingering concerns regarding evaluator-method coupling and the practical efficiency of the 1.4-minute iterative video generation loop, the authors provided compelling mitigations during the rebuttal that satisfied the remaining reviewers. Specifically, the authors demonstrated that their SCPE framework is backbone-agnostic by substituting the VLM agents with an open-source model (Qwen 3.5) while maintaining high performance, and they validated their HOI-Eval metric by showing a strong 0.60 Pearson correlation with human judgment. Furthermore, while the computational latency makes it currently unsuitable for real-time editing, the community will still benefit significantly from its utility as a zero-shot, training-free oracle for generating high-fidelity multimodal data.

Given the much-needed paradigm shift introduced by the HOI-Edit benchmark—which evaluates complex interactions across cognitive hierarchies rather than static pixel modifications—alongside the highly effective ReCamMaster integration that resolves spatial shifting bottlenecks, the core technical contributions are solid.